# Mutant p53s generate pro-invasive niches by influencing exosome podocalyxin levels

David Novo [1], Nikki Heath[1], Louise Mitchell[1], Giuseppina Caligiuri[2], Amanda MacFarlane[1], Dide Reijmer[1], Laura Charlton[3], John Knight[1], Monika Calka[2], Ewan McGhee[1], Emmanuel Dornier [1], David Sumpton [1], Susan Mason[1], Arnaud Echard [4], Kerstin Klinkert[4], Judith Secklehner[1,5], Flore Kruiswijk[1], Karen Vousden[1,2,7], Iain R. Macpherson [1,2], Karen Blyth [1,2], Peter Bailey[6], Huabing Yin [3], Leo M. Carlin [1,5], Jennifer Morton[1,2], Sara Zanivan [1,2] & Jim C. Norman[1,2]

Mutant p53s (mutp53) increase cancer invasiveness by upregulating Rab-coupling protein (RCP) and diacylglycerol kinase-α (DGKα)-dependent endosomal recycling. Here we report that mutp53-expressing tumour cells produce exosomes that mediate intercellular transfer of mutp53's invasive/migratory gain-of-function by increasing RCP-dependent integrin recycling in other tumour cells. This process depends on mutp53's ability to control production of the sialomucin, podocalyxin, and activity of the Rab35 GTPase which interacts with podocalyxin to influence its sorting to exosomes. Exosomes from mutp53-expressing tumour cells also influence integrin trafficking in normal fibroblasts to promote deposition of a highly pro-invasive extracellular matrix (ECM), and quantitative second harmonic generation microscopy indicates that this ECM displays a characteristic orthogonal morphology. The lung ECM of mice possessing mutp53-driven pancreatic adenocarcinomas also displays increased orthogonal characteristics which precedes metastasis, indicating that mutp53 can influence the microenvironment in distant organs in a way that can support invasive growth.

[1] Beatson Institute for Cancer Research, Glasgow G61 1BD Scotland, UK. [2] Institute of Cancer Sciences, University of Glasgow, Glasgow G61 1QH, UK. [3] School of Engineering, University of Glasgow, Glasgow G12 8LT, UK. [4] Membrane Traffic and Cell Division Lab, Cell Biology and Infection Department, Institut Pasteur, 25-28 rue du Dr Roux, Paris 75724, France. [5] Inflammation, Repair & Development, National Heart & Lung Institute, Imperial College London, London SW7 2AZ, UK. [6] Wolfson Wohl Cancer Research Centre, Institute of Cancer Sciences, University of Glasgow, Glasgow G611QH, UK. [7] Present address: Francis Crick Institute, 1 Midland Road, London NW1 1ST, UK. These authors contributed equally: David Novo, Nikki Heath. Correspondence and requests for materials should be addressed to J.C.N. (email: j.norman@beatson.gla.ac.uk)

Loss of wild-type p53 function is a key watershed in tumour initiation and progression. This occurs through loss of p53 expression or mutations that generate p53 proteins defective in wild-type function. A gain-of-function for mutant p53 (ref. [1]) (mutp53) first became apparent following the construction of a mouse model of Li-Fraumeni syndrome[2]. In this animal, wild-type p53 was replaced with mutp53 alleles (p53[R270H] and p53[R172H]) and this led to the spontaneous growth of tumours with more aggressive phenotypes than was observed in p53 null mice. The ability of mutp53 to drive metastasis was then demonstrated using autochthonous mouse models of pancreatic cancer[3], and cells isolated from mutp53 pancreatic tumours are more invasive than their p53 null counterparts[4], indicating that mutp53's pro-metastatic gain-of-function is associated with increased cell migration[5,6].

The way in which integrin receptors for the ECM are trafficked through the endosomal pathway and returned, or recycled, to the plasma membrane is key to the migratory behaviour of cancer cells[7,8]. The Rab11 effector, Rab-coupling protein (RCP), controls integrin recycling, and it is now clear that mutant p53s can drive invasive migration by promoting RCP-dependent integrin recycling[6]. The characteristics of the tumour ECM is closely correlated with disease progression, resistance to therapy, and poor prognosis, and there is now much interest in targeting the ECM and its receptors as an anti-cancer strategy[9]. The ECM within tumours is deposited primarily by fibroblastic cells (carcinoma-associated fibroblasts (CAFs)) and this is controlled by autocrine and paracrine pathways which relay signals between malignant cells and CAFs[10]. Furthermore, ECM proteins are assembled and extensively re-modelled following secretion, and the way that integrins are trafficked through the endosomal system can control this[11,12]. Finally, secreted factors, such as lysyl oxidase, can act directly on the ECM to introduce cross-links which alter ECM organisation and stiffness in way that promotes local invasiveness[13].

The ECM of target organs also contributes to metastasis, and cells in the primary tumour can influence this by releasing factors into the circulation. For instance, lysyl oxidase not only influences the ECM of primary tumours in the breast but also primes bone marrow niches to enable metastatic seeding[14]. Primary tumours also prime metastatic niches by releasing extracellular vesicles (EVs)—such as exosomes—into the circulation. Exosomes released by melanomas can influence differentiation of bone marrow-derived stem cells to promote their mobilisation to tissues—such as the lung—where they contribute to deposition of ECM proteins[15]. More recently exosomes from pancreatic adenocarcinoma cells were shown to promote TGFβ secretion from Kupffer cells which led to fibronectin production by liver stellate cells[16]. However, despite studies outlining how certain factors, such as oncogenic proteins and microRNAs might be transmitted between cells, the molecular players that mediate the pro-metastatic effects of oncogenes are not yet clear.

Here we report that primary tumours expressing mutp53s with pro-metastatic gain-of-function can evoke pro-invasive alterations to the ECM in a metastatic target organ, and we provide the molecular details of how this occurs.

## Results

**Mutp53 promotes release of diffusible pro-invasive factor(s).** 'Organotypic' plugs of acid-extracted type I collagen in which the ECM has been 'preconditioned' by human fibroblasts recapitulate key characteristics of the stromal microenvironment[17]. When plated onto organotypic plugs preconditioned with telomerase-immortalised human fibroblasts (TIFs), H1229 non-small cell lung carcinoma cells (which do not express p53) (H1299-p53[−/−]) were poorly invasive, with most cells residing in the upper

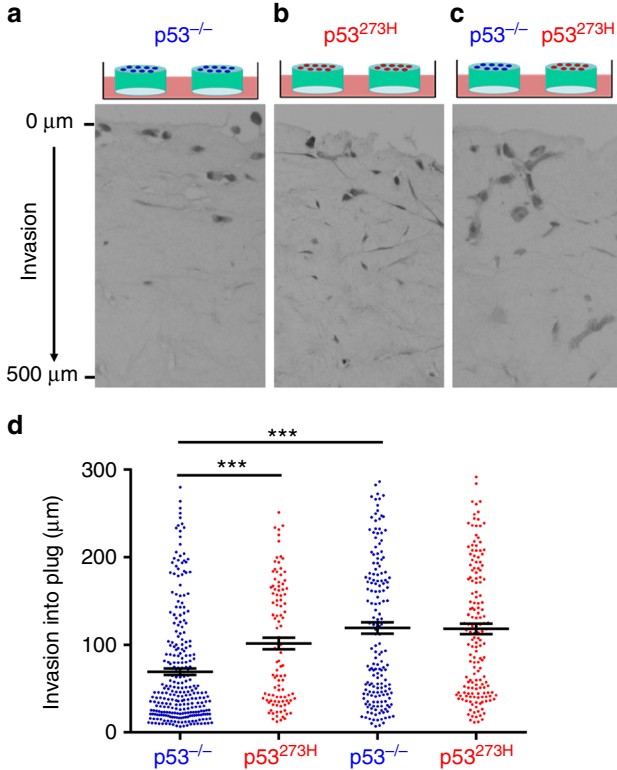

**Fig. 1** Mutant p53 promotes release of diffusible factors to foster tumour cell invasion in an organotypic microenvironment. Organotypic plugs were generated allowing acid-extracted rat tail collagen to polymerise in the presence of telomerase-immortalised human dermal fibroblasts (TIFs). Fibroblast-containing plugs were conditioned for 2 days to allow TIFs to deposit and remodel the ECM. Preconditioned plugs were overlaid with H1299-p53[−/−] (**a**) or H1299-p53[R273H] (**b**) cells and placed onto grids in independent Petri dishes containing culture medium. In **c** plugs which were overlaid with H1299-p53[−/−] were placed onto grids in the same Petri dish as those overlaid with H1299-p53[R273H] cells, thus allowing the possibility of exchange of diffusible factors between plugs. Tumour cells were allowed to invade for 10 days, followed by fixation and visualisation of tumour cells with H&E. The distance between each tumour cell and the top of the plug was determined and plotted in **d**. Bars are mean ± SEM, $n = 8$ plugs; *** $p < 0.001$ Mann–Whitney test

portion of the plug 10 days after plating (Fig. 1a, d). By contrast, H1299 cells expressing the conformational gain-of-function mutant of p53, p53[R273H] (H1299-p53[R273H]) invaded extensively into organotypic plugs (Fig. 1b, d).

Pro-invasive oncogenic pathways can operate in a non-cell autonomous fashion by promoting release of diffusible factors. To test whether mutp53 promotes release of pro-invasive factor(s), we placed organotypic plugs containing H1299-p53[R273H] cells in the same Petri dish as plugs plated with isogenic p53 null cells (H1299-p53[−/−]) (Fig. 1c). When cultured in this way, H1299-p53[−/−] cells displayed invasive behaviour that was indistinguishable from H1299-p53[R273H] cells (Fig. 1c, d). These data indicate that the mutant p53 invasive gain-of-function phenotype may be transferred via diffusible factor(s).

**Mutp53-expressing cells release exosomes to influence RCP-dependent integrin trafficking and cell migration in p53 null cells.** Mutp53-expressing cells migrate faster and more erratically on 2D substrates than their p53 null counterparts[6]. Indeed,

persistence and forward migration index (FMI) of H1299 cells migrating into scratch-wounds is suppressed by expression of mutp53 (Supplementary Figure 1a–c). To represent these changes graphically, we calculated the differences between the persistence and FMI of mutp53-expressing and p53 null cells—these we term the ΔPersistence and ΔFMI—and plotted them as $x$ and $y$ coordinates, respectively (Supplementary Figure 1d).

Conditioned medium from H1299-p53$^{R273H}$ donor cells significantly suppressed the migratory persistence and FMI of H1299-p53$^{-/-}$ recipient cells, and this was opposed by siRNA of p53$^{R273H}$ in the donor cells (Supplementary Figure 1e). Moreover, depletion of exosomes by centrifugation completely opposed the ability of conditioned medium collected from H1299-p53$^{R273H}$ cells to suppress the migratory persistence and FMI of p53$^{-/-}$ H1299 cells, indicating the likelihood that the diffusible factor(s) responsible for transfer of mutant p53's migratory gain-of-function is/are associated with exosomes (Supplementary Figure 1f).

Nanoparticle tracking, sucrose density gradient centrifugation and transmission electron microscopy (TEM) indicated that the abundance, average protein content, size distribution, and density of exosomes released by H1299 cells was not reproducibly altered by expression of mutp53 (Supplementary Figure 2a–e). Moreover, immunogold TEM indicated that the majority of EVs from H1299 cells were CD63 positive and this was not altered by expression of mutant p53 (Supplementary Figure 2f). Furthermore, a number of exosome markers (CD9, CD63, tsg101, HSPA8) did not differ between exosomes released by p53$^{-/-}$ and mutant p53-expressing cells, and p53 itself was not detectable in exosome preparations (Supplementary Figure 2g). Despite these physical similarities, we isolated exosomes from H1299 cells expressing either of two p53 mutants known to drive mutant p53's invasive gain-of-function, p53$^{273H}$ or p53$^{175H}$ (mutp53$^{273H}$ or mutp53$^{175H}$-exosomes, respectively) and p53 null H1299 cells (termed 'p53$^{-/-}$-exosomes') and compared the ability of these to influence receptor recycling in p53 null-recipient H1299 cells. α5β1 integrin, cMET, and the transferrin receptor (TfnR) recycling was significantly increased by pre-incubation of H1299-p53$^{-/-}$ recipient cells with 'mutp53$^{273H}$ or mutp53$^{175H}$, but not—p53$^{-/-}$-exosomes (Fig. 2a). Moreover, use of a DGKα inhibitor (R59022) indicated that this response was dependent on the RCP and DGKα-regulated recycling pathway previously found to be activated in mutp53-expressing cells[8]. By contrast, α5β1, cMET, and TfnR internalisation was not influenced by treatment with mutp53$^{R273H}$-exosomes (Supplementary Figure 3a).

To investigate the ability of exosomes to mediate intercellular transfer of mutant p53-driven migratory characteristics, we incubated p53 null recipient cells with exosomes purified from H1299 donor cells expressing either mutp53$^{R273H}$, mutp53$^{R175H}$ (Fig. 2b) or wild-type p53 under control of a doxycycline-inducible promoter (H1299-p53$^{tetON}$) (Fig. 2c). Exosomes from donor cells that expressed mutant p53s evoked migratory characteristics associated with mutant p53's invasive gain-of-function (i.e. suppression of migratory persistence and FMI and significantly increased migration speed) in p53 null recipients, whereas exosomes from H1299 cells expressing wild-type p53 were ineffective in this regard (Fig. 2b, c). Furthermore, increased recycling (of α5β1 and TfnR) and migratory characteristics associated with mutant p53 may be passed via exosomes to A2780 cells - which express wild-type p53 (Supplementary Figure 3b). Finally, all these exosome-driven alterations to migratory behaviour were opposed by knockdown of RCP or by inhibition of DGKα in the recipient cells (Fig. 2d; Supplementary Figure 4a), but not the donor cells (Supplementary Figure 4c). Titration experiments indicated that $2 \times 10^7$ exosomes/mL were sufficient

to transfer mutant p53's migratory phenotype between cells (Supplementary Figure 3c). Thus, the concentration of exosomes which accumulate in the medium bathing mutant p53-expressing cells (approx. $1 \times 10^9$ particles/mL) is 100-fold more than is required to generate a migratory phenotype in recipient cells. Moreover, these data indicate that exosomes from p53$^{-/-}$ cells cannot influence the migratory phenotype of recipient cells even when used at a 100-fold higher concentrations than is required for p53$^{273H}$-exosomes to evoke increased cell migration.

Taken together, these data indicate that both p53 null and mutant p53-expressing tumour cells release exosomes in similar quantities, but those from mutant p53-expressing cells upregulate RCP and DGKα-dependent receptor recycling in p53 null recipient cells to evoke migratory characteristics associated with mutant p53's invasive gain-of-function.

**Mutp53 controls exosomal podocalyxin levels to drive receptor trafficking and cell migration in p53 null cells.** We proposed that altered exosome composition might be responsible for intercellular transfer of mutp53's migratory gain-of-function. SILAC-based proteomics allowed comparison of exosomes purified from H1299-p53$^{R273H}$ and H1299-p53$^{-/-}$ cells which had been labelled with light and heavy SILAC amino acids respectively. Of the 428 proteins that were unambiguously identified, only 4 of these differed significantly between mutp53-expressing and p53 null cells (Fig. 3a; Supplementary Data 1). Podocalyxin (PODXL), a sialomucin associated with cancer aggressiveness[18], was significantly suppressed in mutp53$^{R273H}$-exosomes, and the ability of both the 175 and 273 H mutants of p53 to suppress exosomal PODXL was confirmed by western blotting (Fig. 3b). Moreover, sucrose density gradients indicated that PODXL precisely co-migrated with CD63 at a density of 1.1–1.15 g/mL indicating that PODXL is integrally associated with exosomes, and exosomal-associated PODXL is suppressed by mutp53 (Fig. 3c). RNAseq indicated that PODXL is the second-most significantly downregulated gene in mutp53-expressing H1299 cells - and qPCR and western blotting confirmed that PODXL mRNA and protein levels were suppressed by mutp53$^{175H}$ or mutp53$^{R273H}$ in H1299 cells (Fig. 3d; Supplementary Figure 5a; Supplementary Data 2). By contrast, induction of wild-type p53 did not affect PODXL levels (Supplementary Figure 5b). We and others have previously shown that mutant p53s exert pro-invasive gain-of-function by associating with and inhibiting p63[5,6]. We, therefore, knocked-down p63 and found that this suppressed PODXL level in H1299 cells to a similar extent as mutp53s (Supplementary Figure 5c). This indicates that PODXL expression is under the control of p63 and mutp53 likely suppresses PODXL levels by interfering with p63 function.

To investigate whether suppression of exosomal PODXL levels underpins the transfer of mutant p53's gain-of-function phenotype, we increased PODXL levels in p53$^{R273H}$-exosomes by expressing PODXL-GFP in H1299-p53$^{R273H}$ cells (Supplementary Figure 5d). This did not influence the quantity or size distribution of mutp53$^{R273H}$-exosomes (Supplementary Figure 5e). However, expression of PODXL-GFP in donor cells significantly reduced the ability of mutp53$^{R273H}$-exosomes to drive receptor (α5β1 and cMET) recycling (Fig. 3e) and cell migration (Fig. 3f). In many reports, it is the presence (not the absence) of PODXL and other sialomucins has been linked to cancer progression[18]. Therefore, we were interested in determining the consequences of further reducing PODXL levels in mutp53-expressing cells. Exosomes from mutp53-expressing PODXL knockdown donor cells (Supplementary Figure 4a) had reduced ability to drive receptor recycling and cell migration in p53 null cells (Fig. 3e, f). By contrast, knockdown of α3β1 integrin

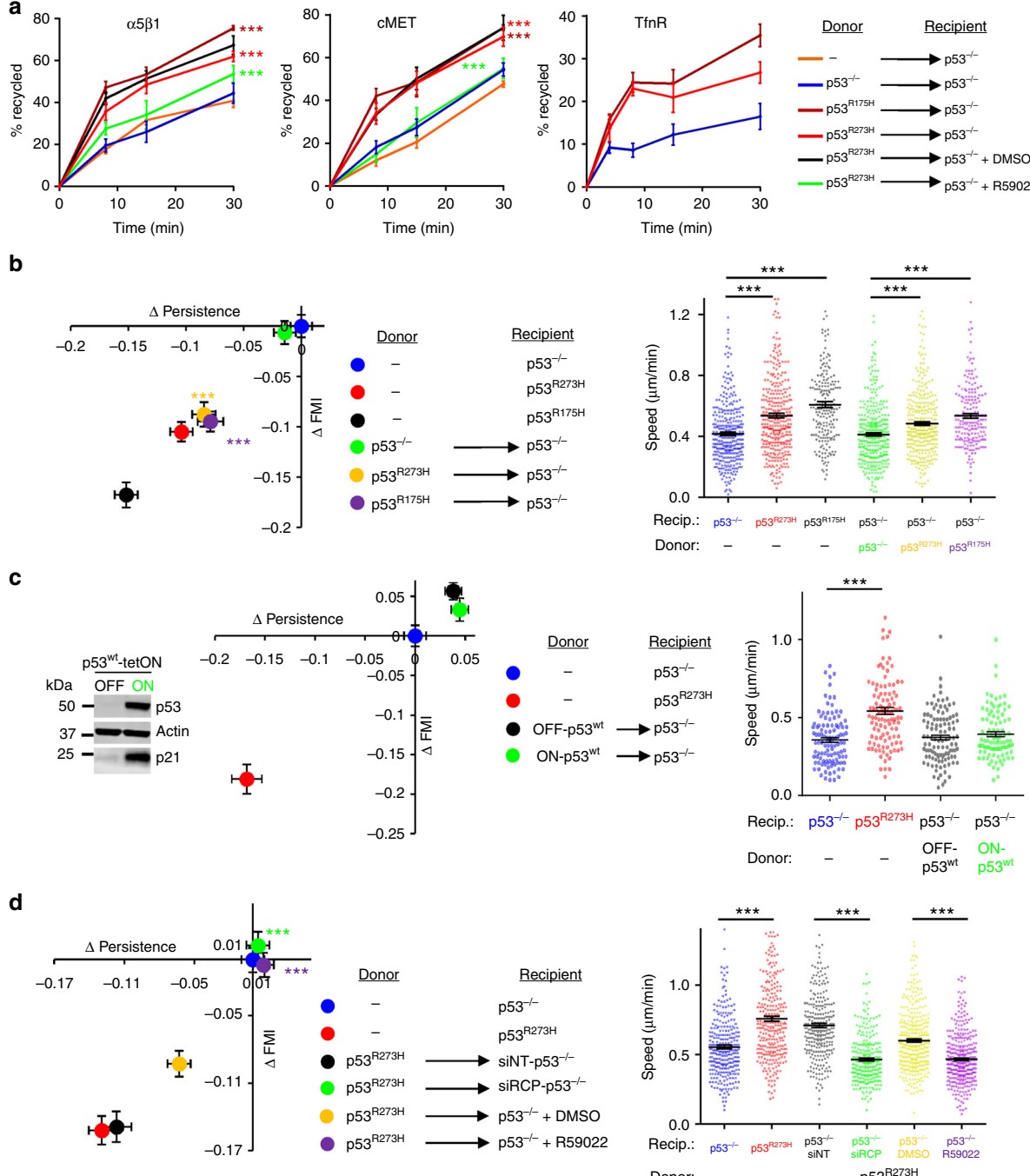

**Fig. 2** Exosomes from mutant p53-expressing cells influence DGKα-dependent integrin trafficking and cell migration in p53 null cells. **a** H1299-p53$^{-/-}$ 'recipient' cells were pre-treated for 72 h with exosomes collected from H1299-p53$^{-/-}$, H1299-p53$^{R273H}$ or H1299-p53$^{R175H}$ 'donor' cells, or were left untreated. Recipient cells were then trypsinised and re-plated. Seventy-two hours following re-plating, recycling of integrin α5β1, cMET and TfnR was determined. Recipient cells were treated with R59022 (10 μM) or DMSO control as indicated. Values are mean ± SEM, $n = 6$; ***red versus blue, and ***green versus black are $p < 0.001$ ANOVA. **b** H1299-p53$^{-/-}$, H1299-p53$^{R273H}$ or H1299-p53$^{R175H}$ 'recipient' cells were pre-treated with exosomes collected from H1299-p53$^{-/-}$, H1299-p53$^{R273H}$ or H1299-p53$^{R175H}$ 'donor' cells, or were left untreated as indicated. Cells were then re-plated and the speed, ΔPersistence and ΔFMI of migration into scratch-wounds determined as for Supplementary Figure 1a–d. Values are mean ± SEM; $n > 195$ cells from three individual experiments; *** in the right panel, and ***yellow versus green and ***purple versus green in the left panel are $p < 0.001$, Mann–Whitney. **c** H1299-p53tetON cells were incubated in the presence or absence of doxycyclin and induction of wild-type p53 was confirmed by western blotting. Exosomes from these cells were incubated with H1299-p53$^{-/-}$ cells, the cells re-plated and the speed, ΔPersistence and ΔFMI of migration into scratch-wounds determined as for **b**. Values are mean ± SEM; $n > 110$ cells; *** are $p < 0.001$, Mann-Whitney. **d** H1299-p53$^{-/-}$ recipient cells were pre-treated with exosomes derived from H1299-p53$^{R273H}$ donor cells. Recipient cells were then transfected with siRNAs targeting RCP (siRCP) or a non-targeting control (siNT), and the characteristics (ΔPersistence, ΔFMI and speed) of their migration into scratch-wounds was determined in the presence and absence of a DGK inhibitor (R59022; 10 μM) or DMSO control. Values are mean ± SEM; $n > 273$ cells; *** in right panel, and ***green versus black and ***purple versus yellow are $p < 0.001$, Mann-Whitney test

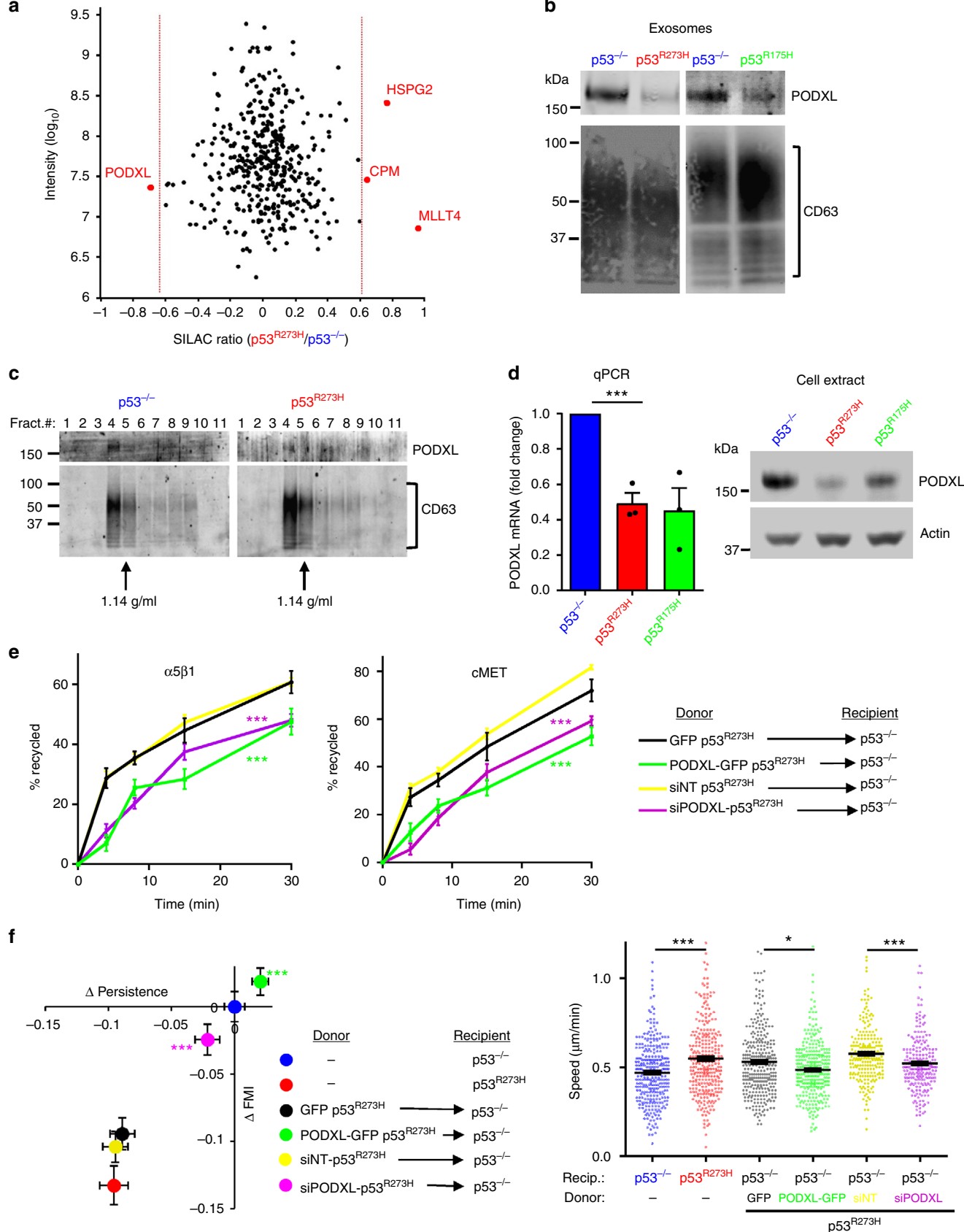

**Fig. 3** Mutant p53 controls exosomal podocalyxin levels to drive receptor trafficking and cell migration in p53 null cells. **a** H1299-p53$^{-/-}$ and H1299-p53$^{R273H}$ cells were SILAC-labelled with heavy and light amino acids respectively. Conditioned media were collected from labelled cells, exosomes purified from these using differential centrifugation, and their proteome analysed by mass spectrometry. Scatter plot indicates the SILAC ratio H1299-p53$^{R273H}$/H1299-p53$^{-/-}$ (Log$_2$ scale) of each protein identified in the exosomal proteome. Proteins to the left and right of the red dotted lines are significantly down- and up-regulated respectively in exosomes from H1299-p53$^{R273H}$ cells (Significance B statistic test, false discovery rate of 5%, Perseus software). These data are extracted from the table presented in Supplementary Data 1. **b** Exosomes from H1299-p53$^{-/-}$, H1299-p53$^{R273H}$, and H1299-p53$^{R175H}$ cells were analysed by western blotting with an antibody recognising PODXL. CD63 was used as sample control. **c** Exosome pellets from H1299-p53$^{-/-}$ and H1299-p53$^{R273H}$ cells were characterised using sucrose density gradient centrifugation followed by western blotting for PODXL and the exosome marker, CD63. **d** H1299-p53$^{-/-}$, H1299-p53$^{R273H}$ and H1299-p53$^{R175H}$ cells were lysed and assayed for the levels of mRNA encoding PODXL (left panel) using qPCR, and for PODXL protein using western blotting (right panel). Values in the left panel are mean ± SEM, $n = 3$; *** is $p < 0.001$ unpaired $t$-test. **e**, **f** Donor H1299-p53$^{R273H}$ cells were transfected with GFP, or PODXL-GFP, siRNAs targeting PODXL (siPODXL) or non-targeting control (siNT). Exosomes collected from these cells were used to treat H1299-p53$^{-/-}$ recipient cells for 72 h before the cells were re-plated and recycling of α5β1 and cMET (**e**) and migratory characteristics of these cells into scratch-wounds (**f**) were determined as for Fig. 2a, b. Values are mean ± SEM. $N > 317$ cells; ***green versus black, and ***purple versus yellow are $p < 0.001$, Mann–Whitney test. In the right panel of **e**, *** is $p < 0.001$ and * is $p < 0.05$, Mann–Whitney

(ITGA3) (Supplementary Figure 4a)—the most abundant exosomal cargo—but whose levels do not differ between mutp53$^{R273H}$ and p53$^{-/-}$-exosomes (Supplementary Data 1) did not oppose the ability of mutp53$^{R273H}$-exosomes to drive migration of recipient cells (Supplementary Figure 6a).

Taken together these data indicate that PODXL is required for exosomes to influence receptor trafficking, but that its levels must be within a certain range for this to occur. The role of mutant p53 is to drive transcriptional suppression of PODXL expression which reduces the exosomal content of this sialomucin to within this range, thus driving RCP-dependent receptor trafficking in recipient cells and allowing transfer of mutp53's gain-of-function to p53 null cells.

**Rab35 associates with PODXL to influence its sorting to exosomes**. PODXL binds to the Rab35 GTPase, and this association controls PODXL trafficking to the plasma membrane[19,20]. Rab35 and PODXL co-immunoprecipitated (Fig. 4a, e) to an extent that is commensurate with the expression levels of PODXL in p53$^{-/-}$ and mutp53-expressing cells respectively (Fig. 4a). Furthermore, knockdown of Rab35 (using SMARTPool siRNAs, an individual siRNA oligo or CRISPR gene editing) reduced PODXL at the cell surface (Fig. 4b; Supplementary Figure 6b, c). Interestingly, Rab35 knockdown led to accumulation of PODXL in CD63-positive late endosomes and, consistently, increased levels of PODXL in exosomes—while not affecting the number and size of exosomes released by H1299 cells (Fig. 4c, d). By contrast, knockdown of Rab27a and Rab27b did not influence the sorting of PODXL into exosomes (Fig. 4c). Mutation of residues in the juxtamembrane region of PODXL's cytoplasmic tail, that have previously been found to be important for Rab35–PODXL association (Val$^{496}$ and Tyr$^{500}$), reduced their co-immunoprecipitation and led to increased trafficking of PODXL to CD63-positive late endosomes and exosomes (Fig. 4e–g). Consistently, the ability of H1299-p53$^{R273H}$-exosomes to influence recipient cell migration was completely opposed by Rab35 knockdown, whereas Rab27 knockdown (which does not influence sorting of PODXL to exosomes) was ineffective in this regard (Fig. 4h). Taken together, these data indicate that because Rab35 (but not Rab27) is required to transport PODXL to the plasma membrane, suppression of Rab35 diverts PODXL to late endosomes, thus increasing exosomal PODXL levels and disturbing exosome-mediated transfer of mutp53's migratory gain-of-function to recipient cells.

**mutp53$^{R273H}$-exosomes promote integrin recycling in fibroblasts to influence ECM architecture**. Treatment with

mutp53$^{R273H}$-exosomes did not significantly increase the invasiveness of p53 null tumour cells in fibroblast-free Matrigel plugs (Supplementary Figure 7a). This raised the possibility that the fibroblasts in the organotypic plugs may contribute to transfer of mutp53's invasive gain-of-function. Indeed, pre-treatment with exosomes from mutant p53-expressing H1299 cells (either H1299-p53$^{R273H}$ or H1288-p53$^{R175H}$) potently increased recycling (of α5β1 and TfnR) in TIFs, and this was opposed by inhibition of DGKα (Fig. 5a). Consistently, pre-treatment of TIFs with mutp53$^{R273H}$ or mutp53$^{R175H}$ exosomes increased their migration speed, decreased migratory persistence and FMI in scratch-wound assays (Fig. 5b), and increased random migration speed of subconfluent TIFs (Supplementary Figure 7b). Moreover, exosome-mediated transfer of these migratory characteristics to fibroblasts was opposed by CRISPR-mediated disruption of Rab35 or PODXL (Supplementary Figure 4b) or overexpression of PODXL-GFP in H1299-p53$^{R273H}$ donor cells (Supplementary Figure 5d), indicating that control of PODXL levels in tumour cells is required for the transfer of mutant p53's migratory phenotype to fibroblasts (Fig. 5b).

To test whether altered DGKα-dependent integrin trafficking might influence ECM deposition, we allowed TIFs that had been pre-treated with mutp53$^{R273H}$-, mutp53$^{R175H}$- or p53$^{-/-}$-exosomes to deposit ECM for 8 days. Immunofluorescence indicated that ECM deposited by fibroblasts is normally organised into bundles of largely parallel filaments, and pre-incubation with p53$^{-/-}$-exosomes did not alter this (Fig. 5c). By contrast, pre-incubation with mutp53 (R273H or R175H) exosomes led to a more branched, orthogonal ECM network. To quantify this, we used grey level co-occurrence matrix (GLCM) analysis. This approach determines the probability (intensity correlation) that pixels at increasing distances (comparison distance) from a given point can be found to have similar intensities. Thus if an image consists mainly of long straight fibres it is possible to travel some distance in a straight line away from a given point without much alteration to intensity, and this will be reflected by long comparison distances for a given intensity correlation—i.e. a long mean decay distance. However, if an image is comprised mainly of short, orthogonally arrayed filaments then the correlation will fall more quickly as one travels away from a given point—yielding a shorter mean decay distance. The intensity correlation of ECM deposited by fibroblasts treated with mutp53$^{R273H}$ or mutp53$^{R175H}$- exosomes decreased more quickly with distance than it did in ECM from untreated fibroblasts or those incubated with p53$^{-/-}$-exosomes (Fig. 5c). Furthermore, inhibition of DGKα (during the ECM deposition period) opposed deposition of orthogonal ECM with a short mean decay distance. Finally, second harmonic generation

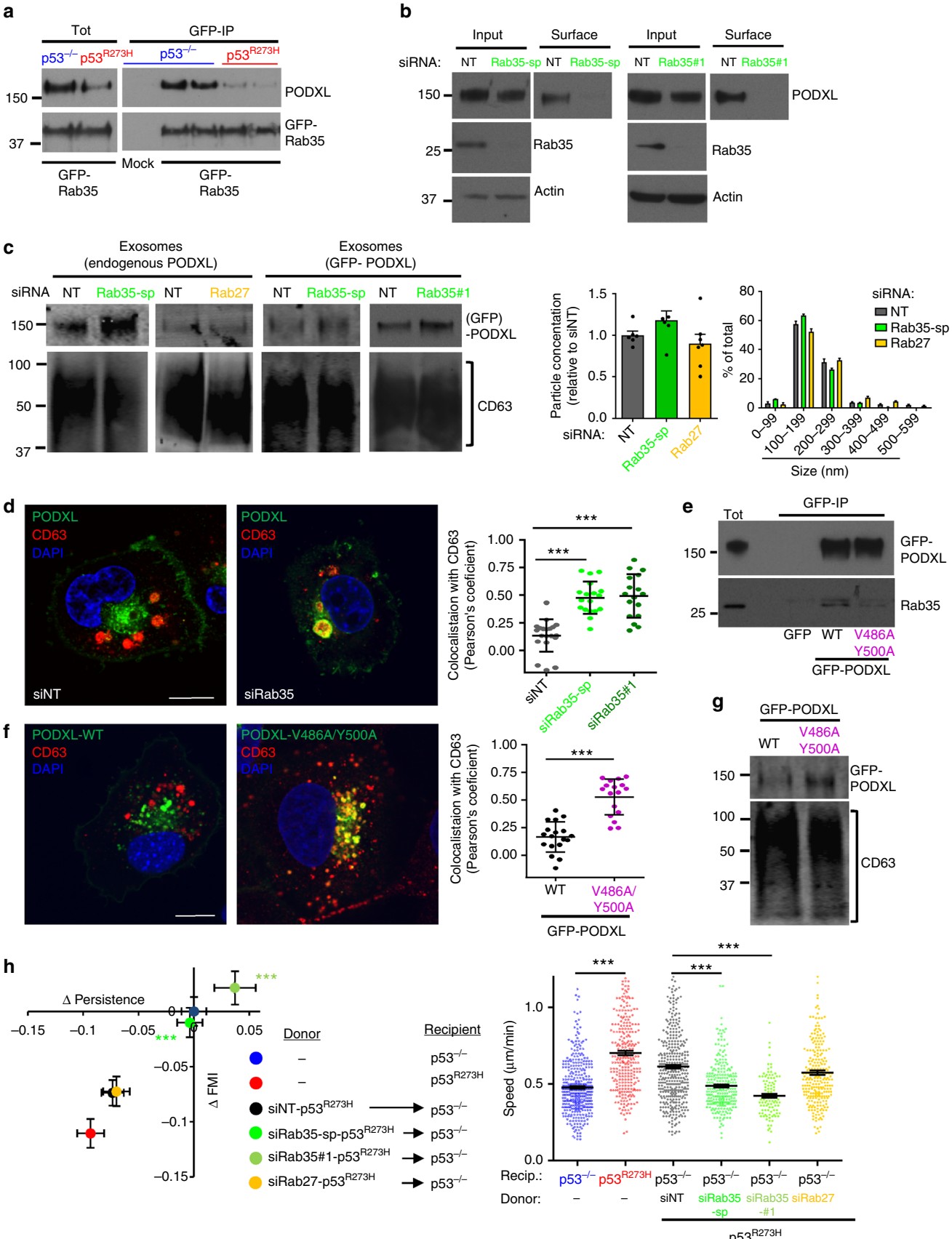

**Fig. 4** Rab35 interacts with PODXL to influence its sorting to exosomes. **a** H1299 (p53$^{-/-}$ or p53$^{R273H}$) cells were transfected with GFP-Rab35 or mock transfected. GFP-tagged proteins were immunoprecipitated using an antibody recognising GFP conjugated to magnetic beads. Rab35 and PODXL were detected in the lysates (Tot) and immunoprecipitates (GFP-IP) using western blotting. **b** H1299-p53$^{R273H}$ cells were transfected with siRNAs targeting Rab35 (SMARTPool (Rab35-sp) or an individual siRNA (Rab35#1)) or a non-targeting control (NT). Cell surface proteins were labelled with NHS-Biotin at 4 °C and precipitated using streptavidin beads. Labelled (surface) and total (input) PODXL were then visualised by western blotting with actin as sample control. **c** H1299-p53$^{R273H}$ cells were transfected with siRNAs targeting Rab35 (Rab35-sp or Rab35#1), Rab27a/Rab27b (Rab27) or a non-targeting control (NT) ± GFP-PODXL. Exosomes were purified by differential centrifugation. Western blotting was used to determine exosomal levels of PODXL and GFP-PODXL with CD63 as sample control. Nanoparticle tracking was used to characterise exosomes (right panels), values are mean ± SEM, $n = 6$ movies from two individual experiments. **d** H1299-p53$^{R273H}$ cells were transfected with siRab35-sp, siRab35#1, or siNT. Cells were fixed and PODXL (green) and CD63 (red) were visualised by immunofluorescence. Bar, 15 μm. ImageJ was used to quantify co-localised pixels as determined by the Costes method. Values are mean ± SEM. $n > 16$ cells. *** is $p < 0.001$, Mann–Whitney. **e** H1299-p53$^{R273H}$ cells were transfected with GFP, GFP-PODXL or GFP-PODXL$^{V486A/Y500A}$. GFP-tagged proteins were immunoprecipitated and Rab35 and GFP-PODXL were detected in the immunoprecipitates as for **a**. **f** H1299-p53$^{R273H}$ cells were transfected with GFP-PODXL or GFP-PODXL$^{V486A/Y500A}$. Cells were fixed and PODXL (green) and CD63 (red) were visualised by immunofluorescence. Bar, 15 μm. Colocalisation was determined as for **d**. Values are mean ± SEM. $n > 16$ cells. *** is $p < 0.001$, unpaired $t$-test. **g** H1299-p53$^{R273H}$ cells were transfected with GFP-PODXL or GFP-PODXL$^{V486A/Y500A}$. Exosomes were collected by differential centrifugation. Western blotting was used to determine exosomal levels of GFP-PODXL with CD63 as a sample control. **h** H1299-p53$^{R273H}$ cells were transfected with siRab27, siRab35-sp, siRab35#1 or siNT. Exosomes collected from these cells were used to treat H1299-p53$^{-/-}$ cells and the characteristics of their migration into scratch-wounds was determined. Values are mean ± SEM; $n > 262$ cells; for siRab35#1 $n = 100$ cells; ***$p < 0.001$, Mann–Whitney

microscopy (SHG) in combination with GLCM analysis indicated that pre-incubation of fibroblasts with mutp53$^{R273H}$-exosomes prior to seeding them into collagen plugs significantly reduced the mean decay distance of fibrillar collagen within these plugs (Fig. 6b, d). Taken together, these data indicate that mutant p53-expressing tumour cells influences organisation of the ECM in 3D microenvironments by releasing exosomes which alter integrin trafficking in fibroblasts.

**mutp53-exosomes encourage fibroblasts to generate a pro-invasive microenvironment**. Exosome-driven alterations to the organisation of the ECM might be expected to influence its mechanical properties to affect tumour cell invasiveness. AFM analysis indicated that the ECM deposited by mutp53-treated TIFs had similar stiffness to that from untreated fibroblasts and, despite observations that pre-treatment with p53$^{-/-}$ exosomes encouraged the deposition of a slightly stiffer ECM (increased Young's modulus), this was not consistent with altered stiffness being associated with the altered properties of mutp53-fostered ECM (Fig. 5d). We, therefore, continued to use AFM to determine the adhesive properties of the ECM. We attached a silica bead to the tip of the AFM cantilever, allowed this to interact with the ECM for a defined time, and then measured the energy required to remove the bead. The energy required to remove a silica bead from the ECM deposited by untreated TIFs was unchanged by pre-treatment of TIFs with p53$^{-/-}$exosomes (Fig. 5d). However, pre-treatment of TIFs with mutp53-exosomes led to a three- to four-fold reduction in energy necessary to remove a bead from the ECM deposited by these fibroblasts (Fig. 5d). Because of this altered stickiness, we assessed the adhesions formed when cancer cells interacted with these matrices. Quantitative analysis of paxillin distribution indicated that MDA-MB-231 cancer cells assembled significantly fewer and smaller 3D cell:matrix contacts when they were plated into ECM deposited by fibroblasts pre-treated with mutp53-exosomes (Fig. 5e), indicating that cell:ECM contacts structures were less well-established in this less adhesive microenvironment. Therefore, we measured the speed at which tumour cells migrated through de-cellularised ECM deposited by exosome-treated fibroblasts. MDA-MB-231 breast cancer cells migrated significantly more quickly through ECM from fibroblasts that had previously been treated with mutp53$^{R273H}$-exosomes than they did through ECM from untreated fibroblasts or those treated with p53$^{-/-}$-exosomes (Fig. 6a). Furthermore, the ability of exosome-

treated fibroblasts to assemble ECM which supported enhanced tumour cell migration was completely opposed by inhibition of DGKα (during the ECM deposition period). Finally, we incubated TIFs with exosomes and allowed them to pre-condition collagen plugs which were subsequently overlaid with H1299 tumour cells. This indicated that H1299 cells, irrespective of their p53 status, invaded efficiently into collagen plugs that had been conditioned by mutp53$^{R273H}$-exosome-treated TIFs (Fig. 6e). By contrast, pre-treatment of TIFs with p53$^{-/-}$-exosomes did not confer increased invadability to organotypic collagen plugs (Fig. 6e) despite increasing the overall levels of fibrillar collagen (Fig. 6c). Taken together, these data indicate that exosomes from mutant p53-expressing tumour cells influence the way that fibroblasts deposit and remodel the ECM so as to generate a micro-environment highly supportive of tumour cell migration and invasion.

**Mutp53-expressing tumours influence ECM architecture in the lungs via PODXL and Rab35-dependent mechanisms**. Exosomes released by a mutp53-expressing tumour in one organ might engender alteration to ECM organisation in other tissues. To test this, we subcutaneously implanted H1299-p53$^{-/-}$ cells or H1299s expressing mutp53$^{273H}$ or mutp53$^{175H}$ into nude mice. When the tumours had grown to 0.8 cm in diameter, we used SHG microscopy in combination with GLCM analysis to analyse ECM organisation in the lung parenchyma. It is important to note that we were unable to detect any H1299 cells in the lung, indicating that any alterations to the lung ECM were owing to factors (such as exosomes) released by the tumour cells and not migration of tumour cells to the lung. The total quantity of fibrillar collagen in the lung did not differ between non-tumour-bearing mice or those implanted with p53 null or mutp53-expressing tumours. However, the fibrillar collagen in the lung parenchyma of mutp53 tumour-bearing mice appeared to be more punctate and less well-organised into long fibres, and the mean decay distance of collagen filaments (as determined using GLCM) was significantly reduced by comparison with those animals with p53 null tumours or non-tumour-bearing animals (Fig. 7a). Importantly, deletion of either PODXL or Rab35 using CRISPR (Supplementary Figure 4b) opposed the ability of H1299-p53$^{273H}$ cells to alter lung ECM organisation in nude mice (Fig. 7a).

We used the KPC and KP$^{fl}$C autochthonous models of pancreatic ductal adenocarcinoma (PDAC). In these, primary

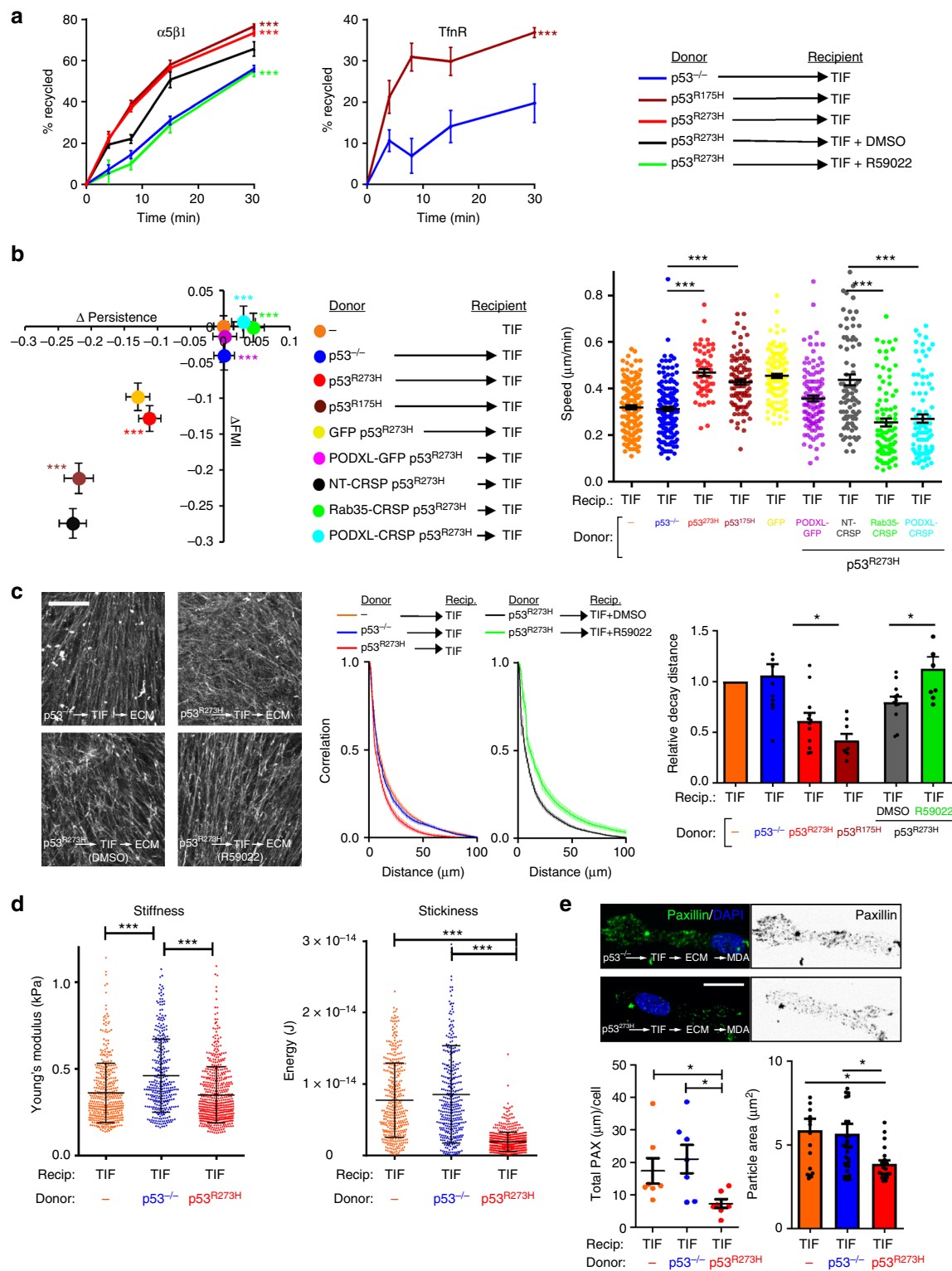

tumour initiation and growth is driven by expression of constitutively-active KRas (LSL-KRas[G12D]) in combination with either deletion of an allele of p53 (p53[fl/+] generating the KP[fl]C mouse) or expression of mutant p53s (either LSL-mutp53[172H] or LSL-mutp53[270H] generating the KP[172]C and KP[270]C mice respectively) under control of the pancreatic-specific Cre recombinase, Pdx-Cre. For comparison, we used KC mice which express LSL-KRas[12G] under control of Pdx-Cre, but the resulting

tumours do not readily progress past the pre-malignant PanIN (pancreatic intraepithelial neoplasm) stage. Importantly for this study, PDAC driven by LSL-KRas and LSL-mutp53[172H] or LSL-mutp53[270H] metastasise to the liver and lung. On the other hand, PDAC driven by KRas in combination with p53 loss (p53[fl/+]) appear with similar penetrance, but do not metastasise[4]. We used SHG followed by GLCM analysis to assess the ECM of the lung parenchyma and perivascular regions at an age (10–12 weeks) in

**Fig. 5** mutp53 exosomes promote integrin recycling in fibroblasts to influence ECM architecture. **a, b** mutp53-expressing R175H or R273H H1299 cells, or cells generated by CRISPR from the latter (PODXL-CRSP; Rab35-CRSP), were transfected with GFP or PODXL-GFP or were left untransfected. Exosomes collected from H1299-p53$^{-/-}$ and the transfected and untransfected mutant p53-expressing H1299 cells were used to treat TIFs and receptor recycling (**a**) and migratory characteristics of these (**b**) were determined as for Fig. 2a, b. In **a** R59022 (10 μM) or DMSO was added to TIFs as indicated. Mean ± SEM, $n = 6$. In **a** ***red versus blue, and ***green versus black are $p < 0.001$, ANOVA. In **b**, $n = >52$; ***red versus blue,***purple versus yellow, ***green versus black, ***light blue versus black are $p < 0.001$, Mann–Whitney. In the right panel of **b**, *** is $p < 0.001$, Mann–Whitney. **c** TIFs were incubated with exosomes from H1299 (p53$^{-/-}$, p53$^{R273H}$, p53$^{R175H}$) cells or left untreated and allowed to deposit ECM in the presence and absence of R59022 (10 μM) or DMSO. ECM was then de-cellularised, stained with antibodies recognising fibronectin and image stacks were collected using confocal microscopy. Extended focus projections of these stacks are displayed in the left panel of **c**, bar, 50 μm. The organisation of the ECM fibres in these was determined using GLCM. The decay curves and the weighted means of the decay distances derived from these are presented in the centre and right panels of **c** respectively. Weighted mean ± SEM, $n = 8$, * is $p < 0.05$, Mann–Whitney. **d** TIFs were treated with exosomes from H1299 (p53$^{-/-}$ or p53$^{R273H}$) cells or were left untreated and allowed to deposit ECM. De-cellularised ECM was analysed using AFM. The left and right panels indicate ECM stiffness and stickiness respectively. Mean ± SEM, $n > 6$ ECM preparations from two individual experiments *** is $p < 0.001$, Mann–Whitney. **e** MDA-MB-231 cells were seeded onto de-cellularised ECM deposited by exosome-treated TIFs as indicated. Cells were fixed and cell:ECM adhesions visualised by immunofluorescence. Top panel shows representative images (bar, 20 μm). Total area of paxillin per cell and average area of paxillin-positive particles are plotted in the bottom left ($n > 6$) and right panels ($n > 16$). Mean ± SEM, * is $p < 0.05$, Mann–Whitney

which primary tumour growth was underway, but metastases were not detectable in the liver or lung. Any animals with premature lung metastases visible by examination of paraffin-embedded, H&E-stained sections were excluded from the analysis —but these were rare. SHG indicated that collagen filaments in the perivascular regions and lung parenchyma of mutp53-driven KP$^{172}$C and KP$^{270}$C animals were shorter and less well-organised, and the mean decay distance (as assessed by GLCM analysis) of these fibres was reduced by comparison with that of KC (Fig. 7b, c) or normal animals. By contrast, for KP$^{fl}$C animals, in which tumours are driven by p53 loss, organisation of lung ECM was indistinguishable from KC animals (Fig. 7b, c). Taken together, these data indicate that hallmark alterations to ECM organisation, which we have established to be driven by the influence of mutp53$^{R273H}$- or mutp53$^{175H}$-exosomes on integrin trafficking in fibroblasts, may also be detected in the lungs of animals bearing autochthonous PDAC expressing the equivalent p53 mutations, but not in the lungs of animals with PDAC with p53 loss.

**Exosomes from mutp53-expressing tumour cells derived from patients with squamous-type PDAC lead to ECM modification.** We determined whether the mutp53 status of cells from human PDAC dictates the capacity of exosomes from these cells to influence ECM deposition. Human PDAC may be categorised into four main subtypes, and these are termed, progenitor, immunogenic, ADEX and squamous[21]. KPC tumours closely recapitulate the characteristics of the squamous subtype of PDAC, so we focussed on the three patient-derived cell lines (PDCLs) from this category. Of these, two (SQ2 and SQ3) expressed mutations that led to ablation of p53 protein expression and were considered, therefore, to be p53 null, while another (SQ1) expressed a mutant of p53 (M237I) with gain-of-function properties (Fig. 8a; Supplementary Figure 7c)[22]. We isolated exosomes from conditioned medium from these PDCLs and incubated them with fibroblasts. We assessed the migration of these exosome-exposed fibroblasts and found that only SQ1 (which expressed a mutant p53 protein) was able to increase migration speed and depress migratory persistence and FMI, as we had previously found for H1299 cells expressing mutant p53s (Fig. 8b). Furthermore, ECM deposited by fibroblasts pre-treated with exosomes from the SQ1 PDCL had reduced mean decay distance (Fig. 8c) and supported increased migration of cancer cells (Fig. 8d). Importantly, we found that knockdown of mutant p53 in SQ1 cells, while not significantly affecting the exosome quantity or size (Fig. 8a), led to increased PODXL levels (Fig. 8a)

and opposed the ability of exosomes released from these cells to support altered fibroblast migration (Fig. 8b) and ECM deposition (Fig. 8c, d).

## Discussion

We have recently reported that conditional deletion of RCP in pancreatic lineages (using Pdx-Cre) opposes metastasis in autochthonous models of mutant p53-expressing PDAC[23]. This is not owing to RCP's ability to control the recycling of α5β1 integrin, but via trafficking of the ephrin receptor, EphA2. However, this does not mean that RCP-mediated integrin trafficking has no role in PDAC metastasis. The progression of tumours from indolent to invasive and onwards to metastasis depends not only on the intrinsic invasiveness and dissemination of cancer cells, but on their ability to influence other cells to generate invasive micro-environments and to prime metastatic niches. Here we show that RCP-regulated α5β1 integrin trafficking is involved in the contribution made by fibroblasts to invasion and metastasis by controlling the way that the ECM is deposited by these cells. Furthermore, we show that mutant p53s expressed by tumour cells activates RCP-dependent integrin trafficking in fibroblasts via an exosome-mediated mechanism.

There are numerous reports that microRNAs and other non-coding RNAs are found within exosomes, and that these can drive processes such as tumour-associated inflammation and metastasis by influencing mRNA processing and translation in target cells[24]. Exosomes from mutant p53-expressing tumours may mediate transfer of a microRNA to influence the polarisation of macrophages to generate an immunosuppressive tumour micro-environment[25]. However, we have conducted extensive quantitative proteomic and RNAseq analyses and cannot demonstrate any alteration to the transcriptome or proteome of target cells following treatment with mutp53-exosomes. This indicates that the influence of exosomes on integrin trafficking does not involve detectable alterations to transcription or translation in target cells and is, therefore, more likely to be mediated by a more direct effect on the endosomal trafficking machinery of target cells. The exosome cargo that is most significantly altered by mutant p53 expression is the sialomucin, PODXL—and this is owing to mutp53's ability to influence PODXL levels in the exosome-producing cells. Alteration of cellular PODXL does not influence the quantity or the size of exosomes released by tumour cells, nor are exosomes with different PODXL content assimilated at different rates by target cells, so it is interesting to speculate how exosomal PODXL levels might influence receptor recycling

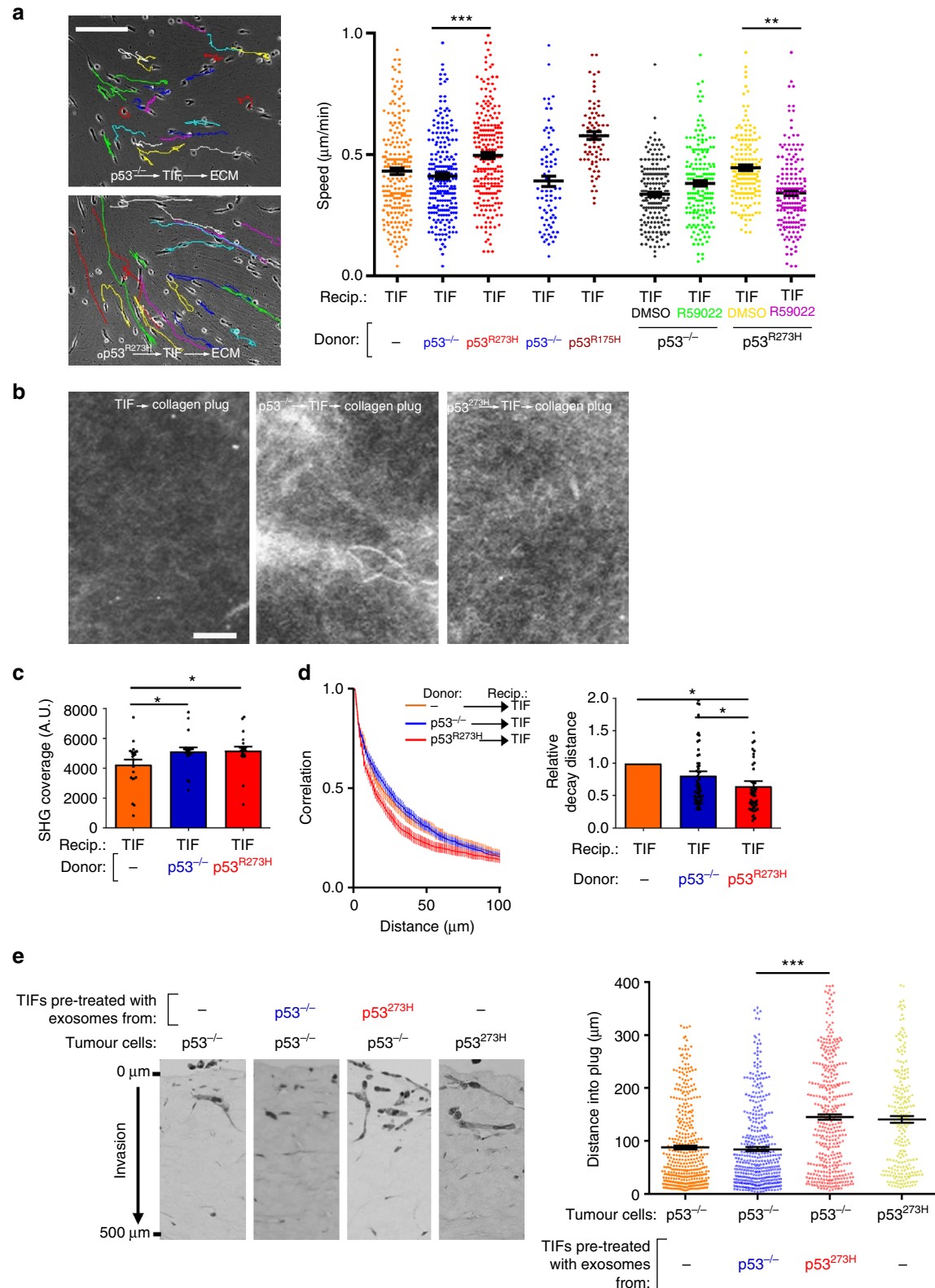

in target cells. PODXL is a glycocalyx component which, by virtue of negative charge imparted by sialylated N- and O- linked oligosaccharide residues, controls the separation of apposed lipid bilayers to promote the opening of lumens during morphogenesis, and to dictate the spacing of kidney podocytes. By respectively controlling PODXL expression and intracellular sorting,

mutant p53 and Rab35 collude to tune levels of exosomal PODXL into a range which is just right for influencing receptor recycling. Any manipulation which places PODXL above or below this range (siRNA of PODXL, overexpression of PODXL, siRNA of Rab35) renders exosomes to be ineffective in driving receptor recycling. Given that sialomucins would be expected to contribute

**Fig. 6** mutp53-exosomes encourage fibroblasts to generate a pro-invasive microenvironment. **a** TIFs were pre-treated with exosomes from H1299-p53$^{-/-}$, H1299-p53$^{R273H}$ or H1299-p53$^{R175H}$ cells and allowed to generate ECM in the presence and absence of R59022 as for Fig. 5c. ECM was then de-cellularised and MDA-MB-231 breast cancer cells plated onto these. The migration of MDA-MB-231 cells through the de-cellularised ECMs was recorded over a 16 h period using time-lapse video microscopy and cell tracking software. Representative tracks are indicated by the coloured lines in the left panels. Bar, 100 μm. The migration speed of the MDA-MB-231 cells was calculated and is presented in the right graph. Values are mean ± SEM, $n > 79$ cells; **$p < 0.01$, ***$p < 0.001$, Mann–Whitney. **b–d** TIFs were incubated with exosomes from H1299-p53$^{-/-}$ (p53$^{-/-}$) or H1299-p53$^{R273H}$ (p53$^{R273}$) cells for 72 h. Exosome pre-treated TIFs were trypsinised, mixed with acid-extracted collagen and the resulting organotypic plugs allowed to polymerise and contract for 3 days prior to collection of image stacks using second harmonic generation (SHG) microscopy. Representative optical slices from these are displayed in **b**. Bar, 4 μm. The coverage of the SHG signal (**c**) and organisation of the fibrillar collagen (**d**) was determined using GLCM as for Fig. 5c. The decay curves from these are presented in the left panels of **d** and the weighted means of the decay distances derived from these curves are displayed in the graph on the right. Values in **c** are mean ± SEM, $n = 22$ fields of view across three separate experiments. Values in **d** are weighted mean ± SEM, $n = 46$ fields of view across three separate experiments; * is $p < 0.05$, Mann–Whitney. **e** Collagen plugs were conditioned for 48 h with untreated TIFs or with TIFs that had been pre-treated for 72 h with exosomes from H1299-p53$^{-/-}$ or H1299-p53$^{R273H}$ cells. Conditioned plugs were overlaid with H1299-p53$^{-/-}$ (p53$^{-/-}$) or H1299-p53$^{R273H}$ (p53$^{R273H}$) cells and these were allowed to invade for 10 days. Plugs were then fixed and tumour cells visualised using H&E. The distance between each tumour cell and the top of the plug was determined using ImageJ. Bars are mean ± SEM, $n > 233$ cells; *** $p < 0.001$ Mann–Whitney test

to the surface charge of exosomes, our data suggest that exosomes within a defined charge range can influence integrin trafficking by acting within the endosomal system, and it will be interesting to determine how other factors which would be expected to influence exosome charge (such as post-translational modification of sialomucins) contribute to the ability of exosomes to interfere with endosomal processes.

CAF activation is associated with increased secretion of fibronectin. More recently, this process has been shown to be promoted by exosomes. Exosomes from PDAC cell lines can bind to liver macrophages to promote release of TGFβ leading to fibroblast activation and fibronectin deposition to prime pre-metastatic niche formation in the liver[16]. However, the alterations in collagen organisation that we find to be driven by mutp53-exosomes are not associated with events that accompany fibroblast activation, such as increased α-smooth muscle actin expression and fibronectin deposition. Rather mutp53-exosomes upregulate RCP-dependent integrin recycling without altering the profile of gene expression in target cells. Integrin endocytosis and recycling can influence ECM deposition and, as fibroblast migration is influenced by mutp53-exosomes, this is also likely to affect the ECM organisation. Long-term time-lapse experiments indicate that fibroblasts in a confluent monolayer normally migrate within a restricted area and this movement is directionally constrained. By contrast, mutp53-exosome-treated fibroblasts undergo much longer range and directionally adventitious movements, and this behaviour may be what leads to the more branched and disorganised ECM that they deposit. Mutant p53-expressing tumours can influence collagen organisation in the tumour stroma and this is associated with ECM cross-linking and assembly of parallel arrays of collagen fibres[26]. Indeed, SHG/GLCM analysis indicates that the mean decay distance of collagen filaments in the stroma of mutant p53-expressing (KPC) PDAC is significantly increased by comparison with the stromal regions of KfC tumours (Figure S7d). Thus, our findings clearly indicate that ECM alterations evoked at some distance from the primary tumour by mutp53 exosomes are distinct from those observed in the primary tumour. The orthogonal/tangled ECM that is deposited by lung fibroblasts under the influence of mutp53-exosomes may contribute to metastatic niche priming in more than one way—for instance by increasing the ability of circulating tumour cells to extravasate and colonise the lungs and/or by influencing the dormancy of micrometastatic colonies, and further work will determine how this occurs. Recently, it has been found that a p53-driven transcriptional programme supports many of the features of tumour-associated fibroblast behaviour, including ECM deposition which fosters cancer cell migration

and invasion[27]. It will be interesting to determine whether regulation of fibroblast p53 function is associated with exosome-mediated transfer of mutp53-like phenotypes and the activation of fibroblast integrin trafficking, cell migration and ECM remodelling.

It is now clear that primary tumours release factors that influence the physiology of other organs to render them more receptive to metastatic seeding. A number of mechanisms have been shown to contribute to this, including mobilisation of immune cells, release of ECM-modifying enzymes (such as lysyl oxidase) and production of exosomes[14,15,28]. Although ECM modification is key to metastatic niche priming, the cellular mechanism through which tumour-derived exosomes lead to altered ECM deposition are not yet clear. Our findings describe a pathway in which a mutated tumour suppressor operates via a well-characterised gain-of-function mechanism to alter the ECM microenvironment to promote tumour cell invasive behaviour. Indeed, mutp53 and Rab35 collaborate to control the PODXL content of exosomes released by tumours cells, and these exosomes promote RCP/DGKα-dependent trafficking of α5β1 integrin in fibroblasts to influence the organisation and adhesive properties of the ECM that they deposit. Although further work will be necessary to determine how this more orthogonal and less adhesive ECM influences cell migration and invasion, this type of ECM organisation, which may be clearly detected in the lungs of mutant p53 tumour-bearing animals, is more conducive to the metastatic seeding of tumour cells. Thus, we have identified a new mechanism that may drive the morbidity of mutp53-expressing tumours and highlight an intercellular communication pathway consisting of a number of measurable well-characterised components (including Rab35, PODXL, RCP, DGKα, collagen organisation) which are amenable to pharmacological intervention and may constitute viable biomarkers to indicate the presence of metastatic tumours.

## Methods

**Cells, qPCR primers and antibodies**. H1299 (p53$^{-/-}$ and p53$^{R273H}$/$^{R175H}$) (from ATCC) TIFs (in house, Beatson Institute) and MCF7 (ATCC) cells were cultured in Dulbecco's modified Eagle medium (DMEM, Life Technologies) supplemented with 10% fetal bovine serum (FBS) (Gibco), 1 mM L-glutamine, 100 μg/mL streptomycin and 100 U/mL penicillin. For all experiments involving exosomes, exosome-depleted serum was used throughout. PDCLs were generated as previously described[1–4] and cultured in M199/F12 HAM medium (1:1) (Life Technologies) supplemented with 7.5% filtered FBS (Hyclone, Thermo Scientific), 15 mM HEPES (Life Technologies), 2 mM L-glutamine (Life Technologies), 20 ng/mL EGF (Life Technologies), 40 ng/mL hydrocortisone (Sigma), 5 ng/mL apo-Transferrin (Sigma), 0.2 IU/mL Insulin ActRApid (Life Technologies), 0.06% glucose (Sigma), 0.5 pg/mL Tri-iodotyronine (Sigma), 1 × MEM vitamins (Life Technologies) and 2 μg/mL O-phosphoryl ethanolamine (Sigma). All cell lines were

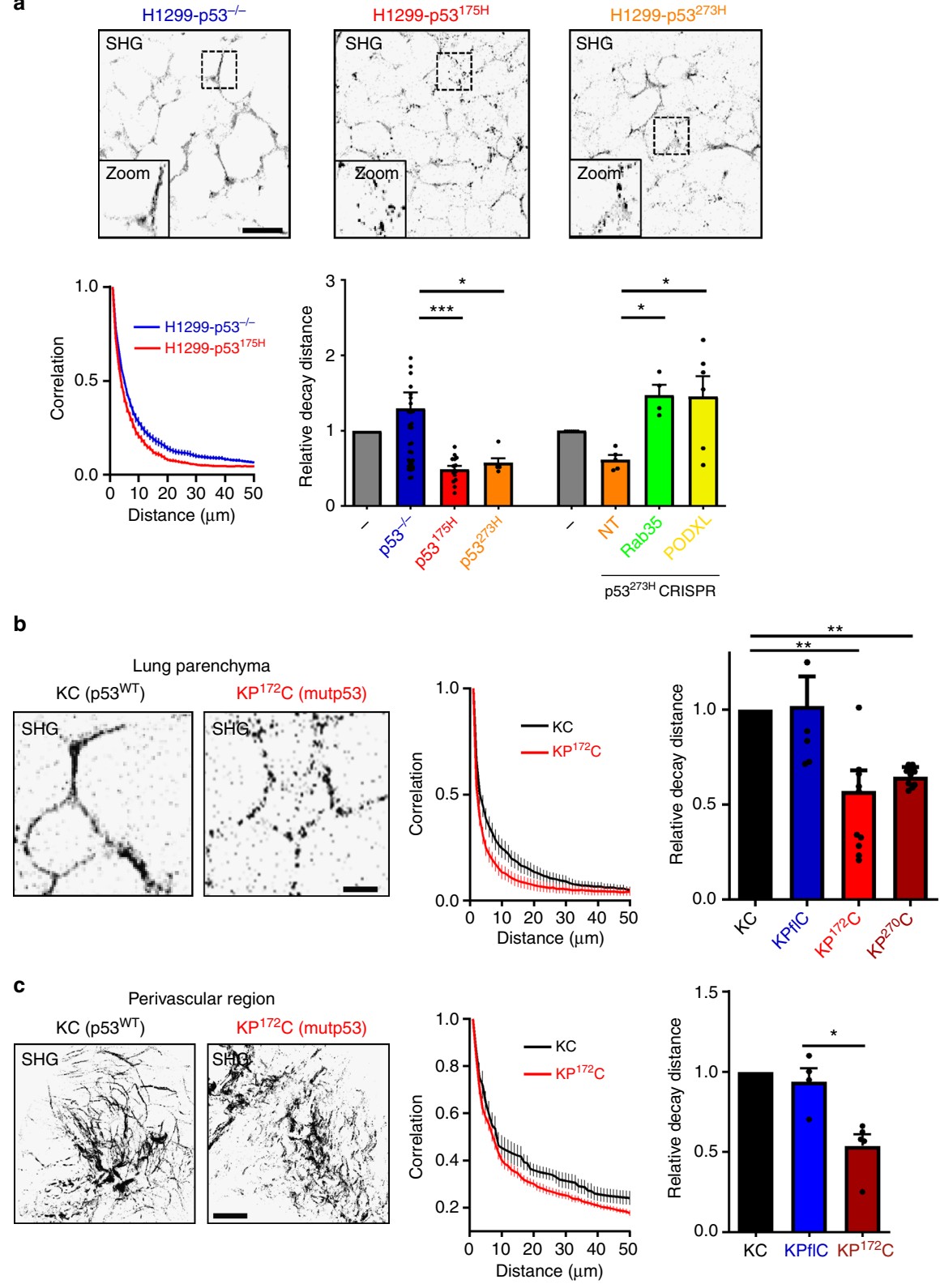

routinely tested for mycoplasma contamination. H1299 cells were transfected using the AMAXA system–Solution V with the X-001 electroporation protocol. PDCLs were transfected using Lipofectamine RNAiMAX (Thermo Scientific) according to the manufacturer's instructions. To suppress p53 the siRNA forward oligo was GACUCCAGUGGUAAUCUACUU, and for p63 it was UGA ACA GCA UGA ACA AGC U (TT). siRNAs for RCP, DGKα, PODXL, Rab35, Rab27a, Rab27b and ITGA3 were ON-TARGET siRNAs from Dharmacon. An individual siRNA was also used to target Rab35 and the sequence of this was

GAUGAUGUGUGCCGAAUAU. Antibodies used were: CD63 (Pelicluster M1544, dilution for WB—1:1000; IF -1:200), p53 (in house, DO-1, 1:10000), Rab35 (Cell Signalling 9690S, 1:1000), RCP (in-house raised against RCP[379–649], 1:1000), PODXL (Abcam 150358, 1:1000), p63 (Abcam ab53039, 1:1000), α5 and α3 integrin (BD Pharmingen), cMET (R&D systems), GFP (Abcam ab6556, 1:1000), Actin (Sigma A2066, 1:3000), TSG101 (GeneTex GTX70255, 1:1000), HSPA8 (Cell signalling 8444S, 1:1000), Integrin β1 (BD Pharmingen 610467, 1:2000), CD9 (Abcam ab92726, 1:10000), DGKa (Protein tech 11547-1AP, 1:500), Rab27 (Abcam

**Fig. 7** Mutant p53-expressing PDAC influences ECM architecture in the lungs. **a** H1299-p53$^{-/-}$, H1299-p53$^{R273H}$ or H1299-p53$^{R175H}$ cells, or H1299-p53$^{R273H}$ cells in which Rab35 or Podocalyxin had been disrupted by CRISPR were injected subcutaneously into CD1 Nude mice. Mice were monitored for tumour growth and culled when tumours reached 0.8 cm diameter. Mice were sacrificed by IP injection of pentobarbital, and lungs were inflated with 2% low melting point agarose which was then allowed to solidify. Agarose-filled lungs were sliced using a vibratome, and parenchymal regions were imaged by SHG microscopy. Representative SHG pictures of lungs from CD1 nude mice transplanted with the respective cell line are displayed (top panels). Bar, 100 µm. Organisation of the fibrillar collagen was determined using GLCM as for Fig. 5c. The decay curves from these are presented in the bottom left panel of **a** and the weighted means of the decay distances derived from decay curves are displayed in the graph on the right. Values are weighted mean ± SEM; $n > 4$ animals per condition (except for H1299-p53$^{R175H}$, where $n = 3$); * is $p < 0.05$, *** is $p < 0.001$, Mann–Whitney. **b, c** KP$^{172}$C (Pdx1-Cre:KrasG12D/+: p53R172H/+) or KP$^{270}$C (Pdx1-Cre:KrasG12D/+:p53R270H/+), KP$^{fl}$C (Pdx1-Cre:KrasG12D/+:p53fl/+) or KC (Pdx1-Cre:KrasG12D/+) mice were sacrificed by IP injection of pentobarbital, and the parenchymal (**b**) and perivascular regions (**c**) of the lungs were and analysed by SHG microscopy as for **a**. Representative SHG pictures of lungs are displayed (left panel). Bar in **b**, 25 µm; bar in **c** 100 µm. Fibrillar collagen organisation was determined using GLCM. The decay curves from these are presented in the centre panels of **b, c**. Weighted means of the decay distances derived from decay curves are displayed in **b, c**, right panels). Values are weighted mean ± SEM. In **b**, $n > 7$ lung fields from three animals per condition (except for KC, where there were 6 animals); * is $p < 0.05$, Mann–Whitney. In **c**, $n = 3$ for KC, $n = 4$ for KP$^{fl}$C and $n = 5$ animals for KPC. * is $p < 0.05$, Mann–Whitney

ab55667, 1:1000), p21 (Cell signalling 2947, 1:1000), Fibronectin (BD Pharmingen 610078, 1:100), Paxillin (BD Pharmingen 610052, 1:100), Integrin α3 (Millipore AB1920, 1:1000). Uncropped blots corresponding to the main figures are presented in (Supplementary Figure 8). qPCR primers for PODXL and GAPDH were from QIAGEN.

For the CRISPR/Cas9 knockout of Rab35 and PODXL we used the lentiCRISPR vector (Addgene plasmid #52961) established by Zhang lab[29]. The guide RNAs sequences we used were GTAGCGAACGTGTCCGGCGT, which was generated by Wang et al.[30] as control, Rab35 guide TTGTCAACGTCAAGCGGTGG and PODXL guide GTGAGGTTCAGGACGAGCTG. The lentiviral constructs were cloned and transduced into H1299 cells as described in ref. [31].

**Exosome purification and characterisation.** Conditioned medium was collected and centrifuged to remove live cells (300g), dead cells (2000g) and finally to remove cell debris and larger lipid membrane fragments (10,000g). Exosomes were then pelleted using a 100,000g centrifugation in a SW32 rotor. The pellet was washed in PBS before a final pelleting centrifugation at 100,000 g, after which exosomes were resuspended in a small volume of PBS. For sucrose density gradient centrifugation, exosome pellets were mixed with 1 mL of a 2.5 M solution of sucrose at the bottom of a 12 mL centrifugation tube. Exosomes were overlaid with 11 layers of sucrose decreasing in concentration (from 2 to 0.4 M sucrose using 20 mM HEPES as the diluent). The gradient was centrifuged at 200,000 g overnight using an SW40 rotor. Exosomes were collected from each gradient fraction by a final centrifugation in PBS at 100,000 g.

Nanoparticle tracking analysis was carried out using the NanoSight LM10 instrument according to the manufacturer's instructions. Exosomes resuspended in 200 µL of filtered PBS were diluted 1:30 in filtered PBS before being introduced into the instrument for measurement.

When incubating recipient cells (H1299, A2780, or TIFs) with exosomes we routinely included these in the medium at a concentration of approx. $1 \times 10^9$ particles/mL.

**Electron microscopy.** Exosomes were fixed in 2% paraformaldehyde (Thermo Scientific Pierce) and subsequently adsorbed onto Formvar carbon coated EM grids overnight at 4 °C. Grids were washed with PBS and treated with 1% glutaraldehyde (Sigma) solution for 5 min. This was followed by eight washes with distilled water. Exosomes were visualised by negative staining, grids were incubated with uranyl oxalate (Polysciences) for 5 min and subsequently methyl cellulose-UA (Sigma) for 10 min at 4 °C. Air dried grids were imaged on a transmission electron microscope FEI Tecnai T20 running at 200 kV using Olympus Soft Imaging System software.

For immunogold staining, adsorbed exosomes were subject to four blocking washes with PBS containing 50 mM glycine after initial adsorption onto grids. A second blocking step was then carried out using PBS containing 5% BSA (Sigma) for 10 min. Exosomes were then exposed to CD63 primary antibody (Pelicluster, 1:200) or mouse IgG1 isotype control antibody (Pierce, 1:200) diluted in PBS containing 1% BSA for 30 min. Grids were washed in PBS containing 0.1% BSA six times for 5 min each. Grids were then incubated with anti-mouse 10 nm protein A-gold conjugate secondary antibodies (Cell Microscopy Centre) for 30 min before eight PBS washes. From this point onward the fixation and negative staining protocol was performed as described above. Images were analysed using ImageJ to determine the exosome size.

**SILAC proteomics.** H1299-p53$^{-/-}$ cells were cultured in heavy SILAC (lysine8, arginine10—Cambridge Isotope Labs) and H1299-p53$^{R273H}$ in light SILAC medium. Conditioned media from these labelled cells were mixed and exosomes isolated by differential centrifugation. The exosome pellet was resuspended in 6 M urea for mass spectrometry analysis. Exosome proteins were reduced (10 mM dithiothreitol), alkylated (55 mM iodoacetamide) and digested (Lys C and trypsin).

Peptides were cleaned using stage tips and re-dissolved in 5% acetonitrile/0.25% formic acid. Protein samples were then applied directly to an Orbitrap Elite (LC-MS). Data were searched and quantified using Swissprot (Human) database and MaxQuant software.

**Receptor trafficking and cell migration.** Recipient cells were cultured in the presence of purified exosomes for 72 h. Following this, cells were trypsinised and washed to remove exosomes, re-plated and grown for 48 h to achieve a confluence of 80–90% prior to conducting receptor recycling assays[32]. H1299-p53$^{-/-}$ cells were incubated in serum-free DMEM, transferred to ice, washed twice in cold PBS and surface-labelled at 4 C with 0.2 mg/mL NHS-SS-biotin (Pierce) in PBS for 30 min. Cells were transferred to serum-free DMEM for 30 min at 37 °C to allow internalisation of tracer. Cells were returned to ice, washed twice with ice-cold PBS and biotin was removed from proteins remaining at the cell surface by reduction with MesNa. The internalised fraction was then chased from the cells by returning them to 37 °C in serum-free DMEM. At the indicated times, cells were returned to ice and biotin removed from recycled proteins by a second reduction with MesNa. The DGK inhibitor (R59022) or DMSO control were added as the receptor internalised, and were maintained during the subsequent recycling period. Biotinylated α5β1, cMET and TfnR were then determined by capture-ELISA using Maxisorp (Nunc) plates coated with antibodies recognising human α5 integrin (BD Pharmingen 555651; 5 µg/mL), cMET (R&D systems AF276; 5 µg/mL)) or TfnR (BD Pharmingen 555534; 5 µg/mL)).

For scratch-wound assays, exosome-treated cells were trypsinised and re-plated for 24 h prior to scratch-wounding and analysis of cell migration.

**De-cellularised ECM and organotypic approaches.** Telomerase-immortalised human dermal fibroblasts (TIFs) were cultured in the presence of purified exosomes for 72 h. To produce de-cellularised ECM, gelatin-coated tissue culture-ware was cross-linked with glutaraldehyde, quenched and equilibrated in DMEM containing 10% FBS. Exosome-treated TIFs were trypsinised and re-plated at near confluence (~$2 \times 10^4$ cells/cm$^2$) and grown for 8 days in DMEM containing 10% FBS and 50 µg/mL ascorbic acid. Matrices were denuded of living cells by incubation with PBS containing 20 mM NH$_4$OH and 0.5% Triton X-100, and DNA residue was removed by incubation with DNaseI[33]. For pre-conditioning of organotypic plugs of rat tail collagen, exosome-treated TIFs were seeded in plugs of rat tail collagen 1 which were allowed to contract in full DMEM (DMEM supplemented with 10% FBS) for 14 days. $4 \times 10^4$ H1299 cells were then plated on top of these plugs and cultured for 2 days. Plugs were then transferred to a metal grid and cultured with full DMEM for 1 week followed by fixation in 4% paraformaldehyde before paraffin embedding. Four micrometer sections were then cut and stained using hematoxylin and eosin[17].

**Atomic force microscopy.** The mechanical properties of the cell-derived ECM were carried out with an Atomic Force Microscope Nanowizard II (JPK Instruments) mounted on an inverted optical microscope (Zeiss Observe) with a cell heater attachment. Force indentation measurements were carried out using an AFM probe (Nanoworld, Arrow TL with a nominal spring constant of 0.03 N/m) attached with a 4.8 µm silica microsphere (Bang Labs) as described previously[34]. Thermal calibrations were performed to determine the spring constant of each cantilever before use. Force spectroscopy measurements were performed on 50 randomised locations on each sample by applying a 3 nN force indentation. The Hertzian spherical model was applied to the approach force–distance curves to deduce the elastic modulus of the ECM using an in-house algorithm written in R. The adhesive properties of the ECM were estimated through analysing the energy required to remove the probe from the matrix, which is the total areas of adhesion peaks in the retraction force–distance curves (JPK data analysis software).

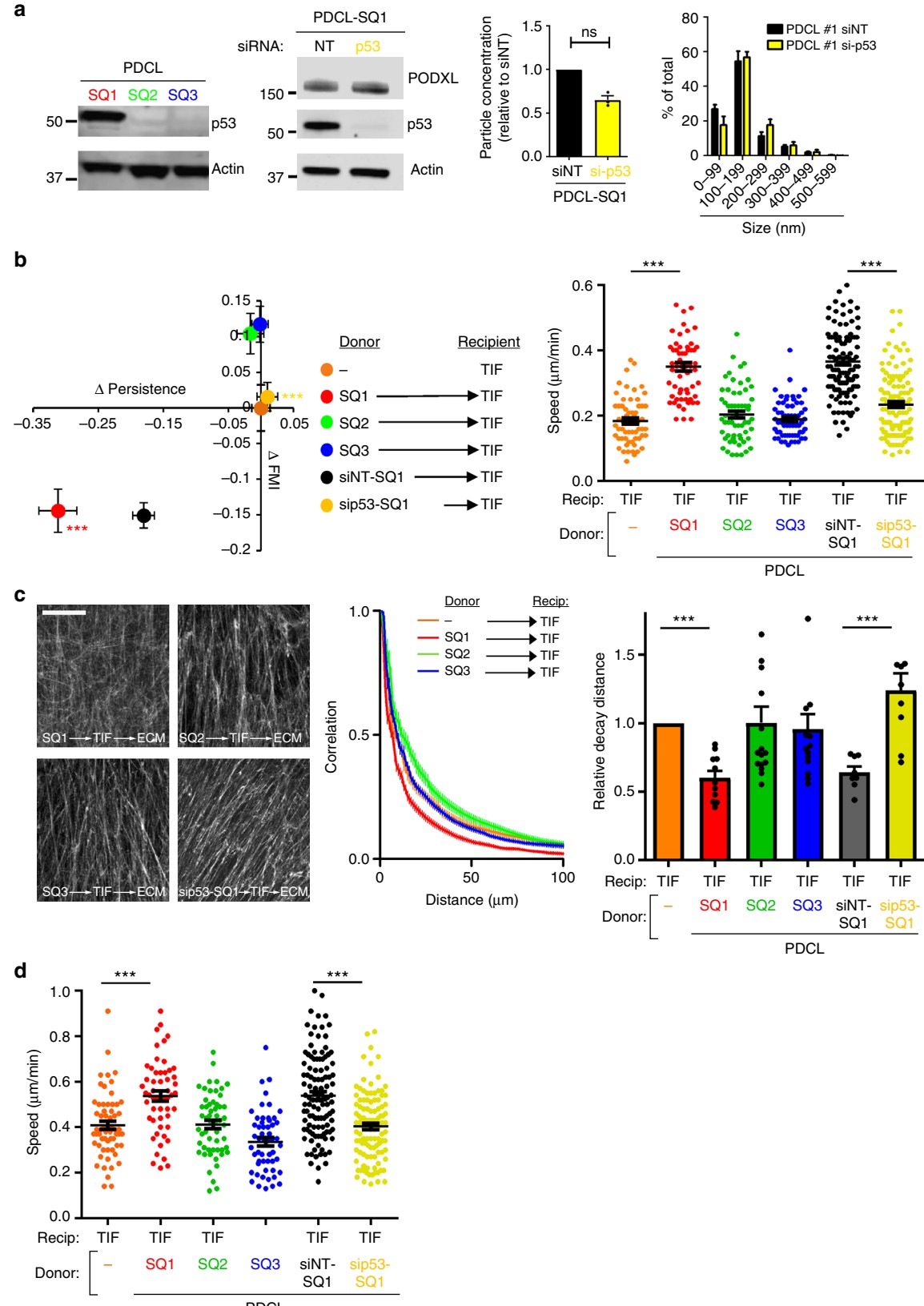

**GLCM analysis of ECM organisation**. Using image sets generated by second harmonic and immunofluorescence imaging, the structure and organisation of the ECM was analysed by applying grey level co-occurrence matrix (GLCM) analysis, a second-order statistical method. Briefly, the intensity of each pixel containing collagen signal is compared to the neighbouring pixels (up to 100 away, corresponding to 100 μm) and a 2D histogram of intensity occurrences compiled, from which statistical parameters of the intensity distribution are calculated such as correlation, homogeneity, contrast and entropy. This has the advantage of removing bias introduced by varying total amounts of signal, changes in the image acquisition and/or signal strength as compared to direct measurements from the raw image data. A bi-exponential model is applied to the correlation decay data and the fit parameters used to calculate a weighted

**Fig. 8** Exosomes released by mutp53-expressing PDCL influence ECM architecture. **a** Levels of p53 in lysates from PDCLs expressing mutp53 – SQ1 – or null for p53 (SQ2 and SQ3) were assessed by western blotting (left panel). SQ1 cells were transfected with siRNAs targeting p53 (si-p53), or a non-targeting control (siNT) and levels of Podocalyxin and p53 in cell lysates were assessed by western blotting, with actin as a sample control (centre panel). Exosomes were collected from transfected cells and purified by differential centrifugation. Nanoparticle tracking was used to determine exosome number and size (right panels). Values are mean ± SEM. $n = 3$. **b** TIFs were left untreated or incubated with exosomes collected from SQ1, SQ2, or SQ3 cells, or from SQ1 cells transfected with si-p53, or siNT. Recipient TIFs were re-plated for a further 48 h and migratory characteristics of these cells into scratch-wounds were determined as for Fig. 2a, b. Values are mean ± SEM. $n > 60$ cells and *** is $p < 0.001$, Mann–Whitney. **c** TIFs were left untreated or incubated with exosomes for 72 h as described above (**b**). TIFs were then trypsinised, re-plated, and cultured for 8 days to allow deposition and remodelling of ECM. ECM was then de-cellularised, stained with antibodies recognising fibronectin and image stacks were collected using confocal microscopy. Extended focus projections of these stacks are displayed in the left panel of **c**, Bar, 50 μm. The organisation of the ECM fibres in these were determined using GLCM as for Fig. 5c. Decay curves from this are presented in the centre panels of **c** and the weighted means of the decay distances are displayed in the graph on the right. Values are weighted mean ± SEM, $n > 8$, *** is $p < 0.001$, Mann–Whitney. **d** De-cellularised ECMs were prepared as described above (**c**) and MDA-MB-231 breast cancer cells were plated onto these. The migration of MDA-MB-231 cells through the de-cellularised ECMs was recorded over a 16 h period using time-lapse video microscopy and cell tracking software. The migration speed of the MDA-MB-231 cells was calculated. Values are mean ± SEM, $n > 53$ cells; ***$p < 0.001$, Mann–Whitney

mean decay distance for use as a parameterisation metric between sample conditions.

**Experimental animals**. KP$^{172}$C (Pdx1-Cre, Kras$^{G12D/+}$, p53$^{R172H/+}$), KP$^{270}$C (Pdx1-Cre, Kras$^{G12D/+}$, p53$^{R270H/+}$), KP$^{fl}$C (Pdx1-Cre, Kras$^{G12D/+}$, p53$^{fl/+}$) and KC (Pdx1-Cre, Kras$^{G12D/+}$) mice (mixed FVB/Bl6 strain) are as previously described[3]. Mice were monitored daily and kept in conventional animal facilities. All animal experiments were performed under UK Home Office licence and approved by the University of Glasgow Animal Welfare and Ethical Review Board. Tumourigenesis was assessed by gross pathology and confirmed by histology. For xenograft experiments, $1 \times 10^6$ H1299 cells were subcutaneously injected into 8-week-female CD1 nude mice. Subcutaneous tumour growth was measured by callipers three times a week until they reached a size endpoint of 8 mm. Mice were sacrificed by intraperitoneal injection of pentobarbital and lungs were inflated with 2% low melting point agarose. Briefly, a small incision was performed in the trachea and liquid agarose was injected with a blunted syringe needle. Mice were then left on ice for 10 min to allow agarose to solidify in lungs. Lungs were dissected and sliced using a vibratome (Campden Instruments 5100mz). Then, either fresh slices of lung were imaged using a Trimscope multiphoton microscope (Lavision) to visualise fibrillar collagen in the parenchyma by SHG, or to visualise fibrillar collagen in the perivascular/peribronchial area by SHG, the lung was fixed with 4% formaldehyde in PBS, sliced as before, mounted in Mowiol with 2.5% DABCO and imaged using an LSM 880 NLO multiphoton microscope (Zeiss).

## Data availability
The data supporting the findings of this study are available within the article and its supplementary information files and from the corresponding author upon request. For the proteomics data, the raw MS files and search/identification files obtained with MaxQuant have been deposited to the ProteomeXchange Consortium via the PRIDE partner repository with the dataset identifier PXD011241.

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

## Acknowledgements

This work was funded by Cancer Research UK and Breast Cancer Now. We acknowledge the Cancer Research UK Glasgow Centre (C596/A18076) and the BSU facilities at the Cancer Research UK Beatson Institute (C596/A17196).

## Author contributions

D.N., N.H., L.M., G.C., A. M.F., D.R., L.C., J. K., M.C., E.M.G., J.S., E.D., L.M.C., F.K., D.S., S.M., J.M., and J.N. performed the experiments and analysed experimental data. A. E., K.V., I.M., K.B., P.B., H.Y., L.M.C., J.M., S.Z., and J.N. supervised the experimental work and/or assisted with the planning of experiments. K.K. and A.E. provided reagents. D.N. and J.N. prepared the figures and wrote the manuscript.

## Additional information

**Competing interests:** The authors declare no competing interests.

