## [Peer Review File · Nature Communications]

Reviewers' comments:

Reviewer #1 (Remarks to the Author):

The manuscript by Heath and Novo et al. illustrates the role of mutant p53 in orchestrating pro-tumorigenic functions via exosomes. The concept of the article is very nice and this reviewer was excited to read the manuscript. However, I was VERY DISAPPOINTED with the overall article as the authors went on a tangent and seldom took it the quality the concept deserves. In my opinion, the manuscript is not of Nature Communications standard and falls behind a long way.

1, The experiments on invasion and migration is a very good starting point. However, the authors went too long with the same concept and similar experiments making the manuscript one dimensional. There is very limited in vivo data and the in vivo data provided is not convincing either. Almost the entire manuscript relies on cell based experiments. The authors fail to mention (even once) the amount of exosomes incubated with the cells. How much exosomes (number or ug/ml) was incubated? This need to be provided in the manuscript, methods and legend. What is the physiological relevance of this part and the concentration used? More strong in vivo data is needed to validate these in vitro findings. At the moment, the manuscript is conceptual/speculative and fails to provide confidence through additional experiments.

2, It was evident that the cargo (RNA and proteins) could be the driving factor for this invasive phenotype. The authors performed SILAC proteomics to identify proteins differentially expressed in exosomes. However, the focus was only on PODXL (downregulated protein in mp53 exos). Have the authors considered other proteins highly expressed in mp53 exosomes?

3, The PODXL link is very weak. As mentioned by the authors, increased expression of PODXL is a prognostic marker for poor survival. Furthermore, various studies have shown that knockdown of PODXL results in attenuation of cancer aggressiveness and migration. Given these findings, the results in the manuscript is contradictory to literature. Can the authors provide data on knocking down PODXL in p53-/- cells and show the p53-/- exos (without PODXL) increases the invasive phenotype?

4, Is p53 in exosomes? Is mp53 in exosomes? No mention of these in the manuscript. What about the validation for the SILAC data? What about the RCP and DGK in exosomes?

5, The data on RAB35 is over interpreted. As implicated by the authors, Rab 27 a and b are important in exosome pathway. However, these proteins are not regulating the biogenesis rather the docking with plasma membrane. Hence, RAB35 may change the contents of exosomes while Rab27 may not. What is connection between PODXL and RAB35? Does PODXL interact with RAB35 when overexpressed in mp53 cells? More mechanistic details are needed or else these data look not detailed and just observations on the periphery.

6, The authors claim that PODXL is a direct target of mp53. What evidence do the authors have to say so other than reduced mRNA expression? Is this a direct regulator or indirect? Does mp53 bind to the promoter of PODXL?

7, The exosomes characterization is not strong in the manuscript. More Western blot for exosomal markers are needed (see Lotval et al, JEV, 2014).

8, The Western blotting in supplementary figure 3a, b and d does not seem to be of publishable quality. This reviewer would like to see the full blot and not squares of one band alone. Given these Westerns on cell lysates are pretty easy, please provide the image without lane cropping.

9, n is missing in supplementary figure legends for error bars.

Reviewer #2 (Remarks to the Author):

This study suggests a gain-of-function role for p53 mutants in cell-to-cell interaction based on extracellular vesicles (EVs) transfer. EVs shed from cancer cells harboring the R273H mutant promoted increased invasiveness and migratory properties. Rab-coupling protein is suggested to

be involved in the integrin recycling levels and fibroblast might be the target cells for the EVs. The authors also identify specific protein cargo for the mutant p53 derived EVs.

In general, I find the concept of this paper is both novel and interesting. The fact that oncogenic alterations in cancer cells can directly affect the tumor microenvironment via released EVs is an important observation which opens many questions and many possibilities, both in diagnostics as well as in therapeutic. I do, however, have several concerns regarding this manuscript in its current state.

1. The study would benefit if some clinical validation would be added. Confirming that these observations are valid in lung cancer patients harboring p53 mutations would strengthen this manuscript.
2. Throughout the entire MS, the authors use one GOF mutant (R273H), which is a central hotspot mutant p53. However, focusing only on one mutant limits the authors ability to create a firm body of findings that supports their conclusions.
3. In the last part of the study, the authors use KPC mice, but a murine model of pancreatic cancer and not lung and also the mutant form is the 172 equivalent to the 175 position in human. While the attempt to show an in-vivo effect is good, this part could be improved. The reasoning to use a different tissue model (pancreas) and a different p53 mutant is not clear and doesn't flow smoothly from previous results. Maybe the authors can use a couple of human pancreatic cell lines harboring R175H to show that relevant phenomena occur with this mutant as well. Also, besides an effect on ECM, no effect is observed either connected to tumor progression or metastases. Also, nothing is tying the observed effect to EV release. This specific effect might be due to a different mechanism.
4. Regarding exosomes isolation – the authors need to provide evidence that PODXL is inside exosomes. All we know, by the data presented, is that PODXL is suppressed by mutp53 in H1299 cancer cells. The authors do not use a filtration steps in their isolation method and we clearly see subsets of larger vesicles (i.e microvesicles). Also, no size-exclusion is conducted to make sure that the proteins (including PODXL) are not pulled down in the UC step but are actually outside the exosomes. Together, this calls for additional proof that this protein is within the exosomes.
5. In the literature, PODXL is proposed to exacerbate cancer progression in most cases including the lung and its overexpression is associated with poor prognosis, while in this study, PODXL suppression is promoting cancer progression. The authors should make this point in the discussion and better yet- they could conduct IHC with lung tumors with or without mutp53 to check PODXL levels and correlate to p53 status and prognosis.
6. Many of the effects are attributed to exosomes, therefore, the authors should describe in detail how many cells are seeded, for how long, what is the confluency etc. Since mutant p53 might also effect proliferation rates, the number of exosomes might differ at the end point of each experiment. As a control, the authors should exclude the possibility that differences in cell number are part of the effect.
7. Was exosome-depleted serum used in the experiments? Exosome-depleted FBS is necessary to exclude any bovine exosome interference. If such sera were not used, the authors should repeat key experiments to verify that the exosomes shed from H1299 are the effectors.
8. The authors focus on fibroblasts as recipient cells; did they also investigate cells in other non-tumor compartments such as immune cells?
 - Consider switching mp53 to mutp53 or any other abbreviation. The term mp53 is sometimes used for mouse p53 and since the authors also show some mice data at the end of the results section, I would try to avoid any confusion.
 - Page 2 line 47 to 50 - provide reference to this section. In general, the manuscript is under referenced.
 - Page 2 line 50 – “A gain-of-function for mutant p53 first became...” many people will argue that the LFS mice are not the “first” evidence but years before that mutants were found to have GOF, and many more will also argue that it is still not proved completely. I would rephrase this sentence.
 - Page 4 line 103 – correct 1229 to 1299.
 - Page 4 line 108 – provide reference to this sentence.
 - In several places throughout the manuscript, a coma is missing (for instance, line 82 – “more

recently...")

Reviewer #3 (Remarks to the Author):

The present work unveils a potentially novel mechanism through which mutant p53 might modify the extracellular milieu of target metastatic organ by influencing the secretion and composition of cancer-released exosomes. These extracellular vesicles might be taken up by normal fibroblasts at distal sites, such as the lung, and alter integrin trafficking and ECM deposition, which displays pro-invasive features.

The work is original and of potentially interest for the wide audience of NC. The notion that p53 mutant modifies the composition of exosomes is novel and a number of well conducted experiments in vitro (in various cell line) and in vivo support this contention. Molecularly, the manuscript is less developed. For example, the identification of Podocalyxin (PDX) as a key component of exosome controlled by mutant p53 is interesting, but it remains unclear whether PDX is necessary and sufficient to render exosome pro-invasive, and how this brought about (how PDX-exosome in particular might affect RCP recycling routes is entirely unclear). Rab35 is also shown to be implicated in PDX-exosome formation but again how this is occurring is unclear. Actually, the data set on RAB35 appear somewhat disconnected with respect to the impact exerted by mutant p53 and unless such a molecular connection is provided, it might be wiser to remove all the finding related to RAB35 that add little in term of advancing our understanding of the mutant p53/exosome connection. These and other issues listed below would need to be addressed to provide the level of molecular insights expected for publication in high-impact journal.

Specific points

Figure 1. in this experiment the invasion of wt, p53^{-/-} and p53R273H mutant cells are tested in organotypic invasion assays. It would be necessary to demonstrate that the same number of cells were initially plated onto the top of the collagen plug. Additionally, H1299-p53^{-/-} cells should be used to restore wt p53. The latter experiment might offer the chance to tests in mixed plugs whether mutant p53 is sufficient to release soluble factors driving the remodeling by dermal fibroblast of the collagen plug.

It must also be clarified how the experiments was performed. In the Legend to this figure it is stated that the plugs were preconditioned for two days with fibroblasts before depositing the various H1299 cells on top of it. This time is likely sufficient to remodel the collagen before the fibroblasts are exposed to any of the soluble cues possibly released by cancerous cells. Is the exposure of the collagen plugs+ Fibroblasts to the pre-conditioned media from p53 mutant cells sufficient to promote invasiveness of p53^{-/-} cells? An experiment of this kind is indeed shown in Figure 6-using exosome preparation.

S1. It is shown that mutant p53 cells appear to migrate at higher speed but reduced persistence. This is apparently in contradiction with a recent finding by the Piel's group showing a universal coupling law between persistence and velocity. To substantiate this remarkable exception to a rather universal law, it would be relevant testing the migration on confined 1D line that greatly facilitate the analysis. It might be sufficient to calculate the instantaneous velocity or mean square displacement and relate this measurement to persistence.

It is also shown that conditioned medium from p53 mutant cells is sufficient to reduced persistence. Is this accompanied by increase in velocity?

Is the migration of TIF affected by exosomes released by mutant p53 cells?

Next, the effect of mutant p53-derived exosomes is analyzed on recycling. The rational to look at recycling is however not obvious as one might expect that a variety of different processes could be targeted and altered by these exosomes and some explanation should be provided.

In Figure 2, it is, in particular, shown that exosomes derived from mutant p53 cells impair alpha5beta1 and cMet recycling in a DGKalpha dependent manner. Is the recycling of other cargo

also perturbed? What about other trafficking routes (e.g internalization)? Stated differently, it is unclear why exosomes should specifically or exclusively target RCP recycling routes. Importantly, information as to nature, quality and features (including EM or AFM analysis) of exosomes should be provided.

Are the mutant p53 cells-derived exosomes generally capable of impacting on recycling of alpha5beta1 and cMET in other cells lines and specifically on A2780?

It is also unclear why increase recycling of different cargo (integrin and cMET) should impact on migration persistence and velocity. Can this effect entirely be understood in term of increased FA turnover? If so, this should be directly shown?

The impact of RAB35 on the ability of mutant p53 exosomes to modulate migratory behaviors of recipient cells is interesting. However, it is unclear whether mutant p53 affect RAB35 expression and more importantly its function? In other terms, what is the relation between mutant p53 and RAB35?

Rab35 has been shown to reduced exosome secretion (Hsu et al. JCB 2010). The author here show, instead, that the number and size of exosomes after RAB35 silencing is not altered. What is the reason of this apparent inconsistency with previously published data?

Please do restore the expression of RAB35 with a siRNA resistant variant or use multiple independent siRNA oligos to assess its impact of this GTPase on exosome-mediated alteration in migratory properties.

The authors also showed that RAB27a and b did not oppose the ability of mutant p53 cell supernatant to perturb migration. One would expect exosome number to be severely affected after this treatment and this should be directly tested. Additionally, if exosome from RAB27a and B KD cells are not affected, then they should not be capable of altering migration as opposed to exosome derived from control mutant p53 cells.

Through Silac-proteomic and RNA-seq, it is shown that PDX is severely reduced in cells and exosomes of p53 mutant cells, but not in p53^{-/-} cells. What is the level of PDX in wt p53 cells and their exosomes?

In addition, it is implied subsequently that the interaction between RAB35 and PDX in co-ip is at the basis of the suppression of migratory properties of mutant p53 cell –derived conditioned media seen after RAB35 silencing. However, it is entirely unclear whether RAB35 loss reduces cells as well exosomal PDX and how this might be occurring.

Also, the set of experiment in Figure 4e and S4c suggest that PDX removal might be sufficient to generate exosomes capable of influencing migratory behavior of recipient cells regardless of p53 mutation status. Is this the case? In other word. is the reduced amount of PDX in exosome sufficient for the exosomal-mediated migratory effects? It should be tested whether exosome purified from wt, p53 mutant, p53^{-/-} in combination with PDX silencing or not are capable of perturbing migration of recipient cells and the recycling of cargos.

In 5b, it is shown that mutant p53-derived exosomes influence the trafficking of RCP cargos and migration also of immortalized dermal fibroblasts. Why a scratch would assays as opposed to a random migration assays was performed? It would be relevant to assess velocity and persistence in randomly moving fibroblasts.

Also, it is stated that “we examined the invasiveness of tumour cells in fibroblast-free Matrigel plugs, we found that treatment with mp53R273H-exosomes conferred only modest and barely-significant increases in the invasiveness of p53 null cells (not shown).”

It would be relevant to show this data.

In 5c-d, it is shown that the ECM derived from fibroblasts exposed exosome-derived from mutant p53 cells is structurally altered with increased orthogonally oriented collagen fibers. It is unclear why the loss of a long and parallel arrays should affect the speed of MDA-MB-231 cells crawling through it. Indeed, looking at the movies, it seems that ECM derived from fibroblasts exposed to p53mutant exosome are organized in parallel arrays that promotes directional migration and

persistent locomotion. Some explanation would seem required to account for the effect reported on migration speeds. Movies where cells are tracked concomitantly with SHG analysis of collagen fibers should be performed.

Finally, It is shown that in pancreatic cancer model driven by mutant p53 the collagen organization in the lung is perturbed. While the results from this analysis are interesting their relationship to exosome released by mutant p53 unclear (for example whether the altered collagen organization in the lungs is caused by exosome released by mutant p53 is not shown). Additionally, it is not clear why the altered ECM, which is show to affect migratory behavior should facilitate the seeding, colonization and growth of metastatic foci at this distal site.

Is the altered ECM a property seen only in Lungs? Is the collagen organization around the primary tumors expressing mutant p53 altered similar to what is seen in the lung? One would reasonably expect that the first targets of mutant p53 exosomes would be the CAFs around the primary tumor? Is this the case? If so, one alternative explanation of the augmented metastatization is that the modified collagen around the tumors might favors local spreading?

Reviewer 1

General comment

The manuscript by Heath and Novo et al. illustrates the role of mutant p53 in orchestrating pro-tumorigenic functions via exosomes. The concept of the article is very nice and this reviewer was excited to read the manuscript. However, I was VERY DISAPPOINTED with the overall article as the authors went on a tangent and seldom took it the quality the concept deserves. In my opinion, the manuscript is not of Nature Communications standard and falls behind a long way.

Our response:

This reviewer questions our approach to the study of exosomal priming of metastatic organs. We, therefore, feel that it is important to state clearly the areas in which our paper makes valuable contributions to this field. There are publications which already describe the phenomenon of exosomal priming of metastatic target organs. However, in our view, the current descriptions of the cellular and molecular mechanisms linking tumour oncogenes to ECM deposition in metastatic target organs are vague. Thus, the purpose of our paper is to use **in vitro/ex vivo approaches to identify and characterise the precise cellular mechanisms linking defined pro-metastatic oncogenic events in tumour cells to ECM deposition by fibroblastic cells, and then to use autochthonous mouse models of cancer to confirm whether the factors controlling these mechanisms operate in vivo**. We have clearly furthered the latter assertion in our revision, and our descriptions of well-characterised exosome-mediated mechanistic links between mutant p53s and ECM deposition, and the molecular players and trafficking events involved in this, make a much-needed contribution to the literature.

1, The experiments on invasion and migration is a very good starting point. However, the authors went too long with the same concept and similar experiments making the manuscript one dimensional. There is very limited in vivo data and the in vivo data provided is not convincing either. Almost the entire manuscript relies on cell based experiments. The authors fail to mention (even once) the amount of exosomes incubated with the cells. How much exosomes (number or ug/ml) was incubated? This need to be provided in the manuscript, methods and legend. What is the physiological relevance of this part and the concentration used? More strong in vivo data is needed to validate these in vitro findings.

We have addressed this as follows:

To address this point, we have strengthened the in vivo data provided in the manuscript. We have now performed a series of experiments in which we have implanted isogenic p53^{-/-} and mutant p53-expressing (p53273H & p52175H) tumour cells as subcutaneous xenografts into nude mice and monitored the influence on ECM deposition in the lung. This approach has allowed us to manipulate the levels of the key mechanistic components of the exosome-mediated pathway that we find to operate in vitro/ex vivo, such as Rab35 and PODXL, in the tumour cells and determine the consequences of this on lung ECM organisation. Thus we have generated mutant p53-expressing H1299

cells in which the genes for Rab35 or PODXL have been deleted using CRISPR. From this we have shown that expression of the key exosomal constituent that mediates transfer of mutant p53's trafficking gain-of-function to fibroblasts (PODXL), and the Rab GTPase (Rab35) which directly controls PODXL sorting into exosomes are required for mutant p53-expressing primary tumours to influence ECM deposition in the lung. These data are now presented in Fig. 7.

We have performed titration experiments to determine the quantity of exosomes which are necessary to evoke alterations to RCP-dependent cell migration in recipient cells. These data are now presented in Fig. S3c and clearly show that concentration of exosomes which accumulate in the medium bathing mutant p53-expressing cells (approx. 1×10^9 particles/ml) is 100-fold more than is required to generate a migratory phenotype in recipient cells. Moreover, these data indicate that exosomes from p53^{-/-} cells are unable to influence the migratory phenotype of recipient cells even when used at a 100-fold higher concentrations than is required for p53^{273H}-exosomes to evoke increased cell migration.

2, It was evident that the cargo (RNA and proteins) could be the driving factor for this invasive phenotype. The authors performed SILAC proteomics to identify proteins differentially expressed in exosomes. However, the focus was only on PODXL (downregulated protein in mp53 exos). Have the authors considered other proteins highly expressed in mp53 exosomes?

We have addressed this as follows:

PODXL is the exosomal protein whose levels are most altered by mutant p53 expression. This is why we have focused on PODXL and found that its levels are indeed a key regulated factor responsible for transferring the mutant p53 gain-of-function phenotype. As this reviewer suggests, we have now determined whether manipulating the levels of other exosomal proteins may also influence the ability of exosomes to influence receptor trafficking and migration of recipient cells. $\alpha 3\beta 1$ integrin is one of the most abundant exosomal cargoes. However, suppression of this integrin in donor cells does not influence the capacity of exosomes from mutant p53-expressing cells to promote migration of recipient cells. These data are presented in Fig. S6.

3, The PODXL link is very weak. As mentioned by the authors, increased expression of PODXL is a prognostic marker for poor survival. Furthermore, various studies have shown that knockdown of PODXL results in attenuation of cancer aggressiveness and migration. Given these findings, the results in the manuscript is contradictory to literature. Can the authors provide data on knocking down PODXL in p53^{-/-} cells and show the p53^{-/-} exos (without PODXL) increases the invasive phenotype?

We have addressed this as follows:

We have now further investigated the influence of exosomal PODXL levels in eliciting recycling, migratory changes and ECM deposition in recipient cells. In addition to the overexpression approach used previously, we have now knocked-down PODXL in mutant p53-expressing donor cells and found that this also reduces the ability of exosomes from these cells to drive recycling and cell migration in recipient cells (Fig. 3e, f). This supports a view that exosomal PODXL is required to transfer the mutant p53 phenotype from cells to cells, but that it must be at an appropriate level to do so. Indeed, it appears that too much or too little PODXL yields exosomes that do not influence integrin trafficking in recipient cells, and that the role of mutant p53 (and Rab35) is to tune the exosomal PODXL levels into a range that does. We have even further extended this analysis by generating H1299 lines that are CRISPR knock-out for PODXL and found that this opposes the ability of mutant p53-expressing cells, when implanted subcutaneously, to modify the lung ECM of recipient animals (Fig. 7).

Finally, we have knocked down PODXL in p53^{-/-} cells, as this reviewer suggests, and found that this does not increase the ability of exosomes from these cells to promote integrin recycling (not shown). Taken together with the data described above, we surmise that this is owing either to the fact that reduction of PODXL levels is not the only p53-controlled factor that allows exosomes to evoke recycling in recipient cells, or that it is not possible to accurately 'tune' PODXL levels using RNA interference into the appropriate range to mimic p53's effects.

4, Is p53 in exosomes? Is mp53 in exosomes? No mention of these in the manuscript. What about the validation for the SILAC data? What about the RCP and DGK in exosomes?

This has been addressed as follows:

p53, RCP and DGK α are not detectable by Western blotting in exosomes from either H1299-p53^{-/-} or H1299-p53R273H cells. These data are now included in Fig. S2g.

5, The data on RAB35 is over interpreted. As implicated by the authors, Rab 27 a and b are important in exosome pathway. However, these proteins are not regulating the biogenesis rather the docking with plasma membrane. Hence, RAB35 may change the contents of exosomes while Rab27 may not. What is connection between PODXL and RAB35? Does PODXL interact with RAB35 when overexpressed in mp53 cells? More mechanistic details are needed or else these data look not detailed and just observations on the periphery.

This has been addressed as follows:

We have now undertaken a detailed mechanistic analysis of the role played by Rab35 in the intracellular sorting PODXL and how this influences its packaging into exosomes. We have confirmed previous data from the Echard lab (Pasteur Institute) and found that Rab35 interacts physically with PODXL and controls its trafficking. We report that Rab35 is required to transport PODXL to the plasma membrane, thus reducing the amount of PODXL available to be delivered to CD63-positive late endosomes and to be

packaged into exosomes. Consistently, knockdown of Rab35 opposes the migration-promoting capacity of exosomes because it diverts PODXL from the plasma membrane and leads to increased packaging of PODXL into exosomes – thus mimicking the situation in p53^{-/-} cells. We have confirmed these results by using a mutant of PODXL which is unable to interact with Rab35 and found that this mutant is packaged into exosomes far more efficiently than wild-type PODXL. By contrast, Rab27 knockdown appears to exert minimal influence on quantity of exosomes released by H1299, and has no effect on the sorting of PODXL into these structures. These data are now presented in Fig. 4.

6, The authors claim that PODXL is a direct target of mp53. What evidence do the authors have to say so other than reduced mRNA expression? Is this a direct regulator or indirect? Does mp53 bind to the promoter of PODXL?

This has been addressed as follows:

We have previously shown that mutant p53 generates its pro-invasive gain-of-function by, at least in part, acting to suppress p63 activity and expression of genes which are controlled by this p53 family member. We have now shown that siRNA of p63 leads to suppression of PODXL levels in p53^{-/-} cells. This indicates the likelihood that PODXL expression is controlled by p63 and that mutant p53 promotes integrin recycling in recipient cells by opposing p63-driven PODXL expression. These data are now presented in Fig. S5c.

7, The exosomes characterization is not strong in the manuscript. More Western blot for exosomal markers are needed

This has been addressed as follows:

In addition to the sucrose density gradient centrifugation analysis and EM that we had previously conducted, we have now Western blotted for a number of additional exosomal markers and these data are presented in Fig. S2g. We have also confirmed the association of PODXL with exosomes (and the ability of mutant p53 to influence this) using sucrose density gradient centrifugation. These data are now presented in Fig. 3c.

8, The Western blotting in supplementary figure 3a, b and d does not seem to be of publishable quality. This reviewer would like to see the full blot and not squares of one band alone. Given these Westerns on cell lysates are pretty easy, please provide the image without lane cropping.

This has been addressed as follows:

We now provide better quality Western blots and have cropped them less (Fig. S4a, b).

9, n is missing in supplementary figure legends for error bars.

These data are now provided.

Reviewer 2

1. The study would benefit if some clinical validation would be added. Confirming that these observations are valid in lung cancer patients harboring p53 mutations would strengthen this manuscript.

This has been addressed as follows:

Since we have used an autochthonous model (KPC model) of pancreatic ductal adenocarcinoma (PDAC), we have now used primary cell cultures derived from tumours from pancreatic cancer patients to obtain some clinically relevant data. Pancreatic cancer has recently been found to fall into 4 basic subtypes – progenitor, ADEX, immunogenic and squamous. The KPC model is thought to recapitulate the squamous subtype of PDAC. We, therefore identified 2 patient-derived cell lines (PDCLs) from the squamous subtype which were null for p53 and one which expressed a mutant of p53 (M237I) which has previously been shown to display gain-of-function properties. We have found that these cells produce exosomes which influence the migration and ECM produced by recipient fibroblasts in a way that is consistent with their p53 status. Moreover, the ability of the mutant p53-expressing PDCL to release exosomes which promote fibroblast migration and the deposition of ‘tangled’ ECM is opposed by siRNA of mutant p53. These data are now presented in Fig. 8.

2. Throughout the entire MS, the authors use one GOF mutant (R273H), which is a central hotspot mutant p53. However, focusing only on one mutant limits the authors ability to create a firm body of findings that supports their conclusions.

This has been addressed as follows:

In addition to studying the p53R273H mutant, we have now analysed the ability of exosomes from H1299 cells expressing another p53 mutation with gain-of-function properties (R175H) to evoke cell migration (Fig. 2b; 5b), receptor recycling (Fig. 2a; 5a), ECM deposition (Fig. 5c) and to influence lung ECM organisation in nude mice (Fig. 7a). We have also augmented our analysis of the role of autochthonous PDAC to alter ECM organisation in the lung. Our previous manuscript deployed the KPC model of PDAC, which is driven by the p53R172H mutation. We now include data from mouse PDAC autochthonously driven by p53R270H, and by deletion of p53 (KfC) as an additional

control (Fig. 7b, c). Finally, we have included data from patient-derived primary cell lines, one of which expresses the p53-M237I mutation which has gain-of-function (Fig. 8).

3. In the last part of the study, the authors use KPC mice, but a murine model of pancreatic cancer and not lung and also the mutant form is the 172 equivalent to the 175 position in human. While the attempt to show an in-vivo effect is good, this part could be improved. The reasoning to use a different tissue model (pancreas) and a different p53 mutant is not clear and doesn't flow smoothly from previous results. Maybe the authors can use a couple of human pancreatic cell lines harboring R175H to show that relevant phenomena occur with this mutant as well. Also, besides an effect on ECM, no effect is observed either connected to tumor progression or metastases. Also, nothing is tying the observed effect to EV release. This specific effect might be due to a different mechanism.

This has been addressed as follows:

We have now performed a number of experiments to strengthen links between the paper's various components. Firstly, we have looked at a number of PDCLs from pancreatic cancer patients exhibiting squamous-like qualities, and shown that a p53 mutation harboured by one of these promotes the release of exosomes which influence ECM production by fibroblasts (Fig. 8). Secondly, we have used two other autochthonous PDAC models to augment the use of the KC and KPC animals. To provide another mutant p53 which is more consistent with the rest of the paper, we have used $KP^{R270H}C$ mouse and report that this leads to similar lung ECM changes as seen in the $KPR^{172H}C$ animals (Fig. 7b, c). Also, because the majority of the paper is predicated on comparisons between p53^{-/-} and mutant p53-expressing cells, we have used a model of PDAC which is driven by KRas in combination with p53 loss – which we term the KflC mouse. KflC mice have similar lung ECM to wild-type mice and the KC controls. These data are now presented in Fig. 7b, c). To answer the final component of this comment, we have attempted to directly inject exosomes from mutant p53-expressing tumours into mice to determine whether this influences lung ECM organisation. However, we have found that intravenous injection of these often leads to death, and because of the nature of our site licence for animal experimentation we are unable to perform the retro-orbital injections which have been successfully been deployed by the Lyden group.

In regard of linking alterations to the ECM with cancer cell behaviour, we have performed atomic force microscopy (AFM) to determine whether this is owing to altered physical properties of the ECM. Although, we originally hypothesised that altered stiffness might be responsible for fostering cell migration, this is clearly not the case. Indeed, we report that there is no consistent difference in stiffness between ECM deposited by fibroblasts exposed to mp53 or p53^{-/-} exosomes. However, an AFM analysis in which we measured the force necessary to remove a silica bead which had been placed on the ECM, indicated that the ECM deposited by fibroblasts exposed to mp53 exosomes is significantly **less** sticky than that deposited by untreated fibroblasts or fibroblasts exposed to p53^{-/-} exosomes. We then plated cells into these matrices and

measured the number and area of adhesive 3D contacts formed and found that these were significantly reduced. Thus, the 'tangled', orthogonal matrix deposited by fibroblasts treated with exosomes from mp53-expressing cells has reduced stickiness leading to looser adhesive interactions with cells engaging with it. This is entirely consistent with the increased ability of cells to move through this ECM. These data are now presented in Fig. 5d.

Finally, we acknowledge that the relationship between the ECM organisation and metastasis is, indeed, a key issue and we are currently investigating this. However, our investigations so far indicate that the answers to these questions are not straightforward. At present, we feel that the altered ECM of the lungs (and the liver) of mice bearing KPC (as opposed to KflC or KC) tumours is more able to support the recruitment of a number of migratory cell types – including cells of the innate immune system, such as neutrophils and macrophages. Moreover, we have some (very) preliminary evidence that components of the 'tangled' ECM are better able to support stem-like behaviour of cancer cells. Thus, because our preliminary investigations indicate that the answer to this reviewer's query involves an extensive investigation into the immune microenvironment of metastatic target organs, cancer stem cell behaviour and the relationship between these and ECM deposition, we feel that addressing these questions lies outwith the scope of the current manuscript.

4. Regarding exosomes isolation – the authors need to provide evidence that PODXL is inside exosomes. All we know, by the data presented, is that PODXL is suppressed by mutp53 in H1299 cancer cells. The authors do not use a filtration steps in their isolation method and we clearly see subsets of larger vesicles (i.e microvesicles). Also, no size-exclusion is conducted to make sure that the proteins (including PODXL) are not pulled down in the UC step but are actually outside the exosomes. Together, this calls for additional proof that this protein is within the exosomes.

This has been addressed as follows:

To confirm that PODXL is integrally associated with exosomes, we have performed sucrose density gradient centrifugation. The exosomes from H1299 cells focus on a sucrose gradient (flotation) at a density of approx.. 1.14 g/l, as indicated by the presence of markers such as CD63 and transmission electron microscopy of fractions from these gradients (Fig. S2e, f). Additional analysis indicates that PODXL is also associated with the exosomes present in this fraction and migrates on the flotation gradient in precisely the same place as CD63. These data are now included in Fig. 3c. Incidentally, we have attempted to perform immunogold-EM to further confirm PODXL's presence in/on exosomes, but have been unable to locate an antibody which is capable of detecting PODXL on EM sections.

5. In the literature, PODXL is proposed to exacerbate cancer progression in most cases including the lung and its overexpression is associated with poor prognosis, while in this study, PODXL suppression is promoting cancer progression. The authors should make this point in the discussion and better yet- they could conduct IHC with lung tumors with or without mutant p53 to check PODXL levels and correlate to p53 status and prognosis.

This has been addressed as follows:

We have now conducted a more in-depth analysis of the role played by exosomal PODXL in evoking aspects of mutant p53's gain-of-function in recipient cells. From this it is clear that either increasing or decreasing the levels of PODXL in exosomes from mutant p53 expressing cells reduces the ability of these exosomes to drive receptor recycling etc. in recipient cells (Fig. 3, f). To pursue this in vivo, we have now shown that CRISPR-mediated knockout of PODXL in mutant p53-expressing H1299 cells ablates their ability to alter lung ECM when implanted into nude mice (Fig. 7a). Thus the relationship between exosomal PODXL levels and the generation of pro-invasive niches is not straightforward. Our interpretation of the data so far is that mutant p53 and Rab35 collude to tune the levels of exosomal PODXL into a 'goldilocks' range which is just right for influencing receptor recycling. Any manipulation which places PODXL either above or below this range (siRNA of PODXL, overexpression of PODXL, siRNA of Rab35) renders exosomes to be ineffective in driving receptor recycling. We updated our discussion to outline these data indicating a situation that is both clear and complex.

6. Many of the effects are attributed to exosomes, therefore, the authors should describe in detail how many cells are seeded, for how long, what is the confluency etc. Since mutant p53 might also effect proliferation rates, the number of exosomes might differ at the end point of each experiment. As a control, the authors should exclude the possibility that differences in cell number are part of the effect.

This has been addressed as follows:

We have performed titration experiments to determine the quantity of exosomes which are necessary to evoke alterations to RCP-dependent cell migration in recipient cells. These data are now presented in Fig. S3c and clearly show that concentrations of 1×10^7 exosomes/ml are necessary to transfer mutant p53's migratory phenotype between cells. Thus the concentration of exosomes which accumulate in the medium bathing mutant p53-expressing cells (approx. 1×10^9 exosomes/ml) is at least 100 fold more than is required to generate a migratory phenotype in recipient cells. Moreover, these data indicate that exosomes from p53^{-/-} cells are **unable** to influence the migratory phenotype of recipient cells even at higher concentrations than we routinely employ, indicating the even quite substantial differences in exosome concentration cannot account for the differences observed between those from p53^{-/-} and mutant p53-expressing cells. Finally, when comparing the properties of exosome released by cells

under various conditions, we always ensure that these cells are at the same confluence both at the commencement and the end of the exosome collection period.

7. Was exosome-depleted serum used in the experiments? Exosome-depleted FBS is necessary to exclude any bovine exosome interference. If such sera were not used, the authors should repeat key experiments to verify that the exosomes shed from H1299 are the effectors.

Yes, exosome-depleted serum has been used throughout. This is now clearly stated in the methods.

8. The authors focus on fibroblasts as recipient cells; did they also investigate cells in other non-tumor compartments such as immune cells?

We are planning to do this. In particular, we are planning to investigate the influence of the types of ECM described in this paper on the behaviour of immune cell populations in the lung. However, we consider this to be outwith the remit of the current manuscript.

Reviewer 3

Figure 1. in this experiment the invasion of wt, p53^{-/-} and p53R273H mutant cells are tested in organotypic invasion assays. It would be necessary to demonstrate that the same number of cells were initially plated onto the top of the collagen plug. Additionally, H1299-p53^{-/-} cells should be used to restore wt p53. The latter experiment might offer the chance to test in mixed plugs whether mutant p53 is sufficient to release soluble factors driving the remodeling by dermal fibroblast of the collagen plug. It must also be clarified how the experiments were performed. In the Legend to this figure it is stated that the plugs were preconditioned for two days with fibroblasts before depositing the various H1299 cells on top of it. This time is likely sufficient to remodel the collagen before the fibroblasts are exposed to any of the soluble cues possibly released by cancerous cells. Is the exposure of the collagen plugs+ Fibroblasts to the pre-conditioned media from p53 mutant cells sufficient to promote invasiveness of p53^{-/-} cells? An experiment of this kind is indeed shown in Figure 6-using exosome preparation.

This has been addressed as follows:

Indeed, the same number of cells were plated onto the top of the plugs. We have outlined this more clearly in the methods section and the figure legend. In addition we have clarified the other methodological issues that this reviewer raises by more careful attention to the wording of the figure legend.

S1. It is shown that mutant p53 cells appear to migrate at higher speed but reduced persistence. This is apparently in contradiction with a recent finding by the Piel's group showing a universal coupling law between persistence and velocity. To substantiate this remarkable exception to a rather universal law, it would be relevant testing the migration on confined 1D line that greatly facilitate the analysis. It might be sufficient to calculate the instantaneous velocity or mean square displacement and relate this measurement to persistence. Is the migration of TIF affected by exosomes released by mutant p53 cells?

We have documented, over the last few years, a number of situations (including expression of mutant p53) in which increased migration velocity is accompanied by reduced persistence. Indeed, we feel that this behaviour is a hallmark of the type of migration evoked by expression of mutant p53. We have described, in previous papers (both from our lab, Pat Caswell's and Johanna Ivaska's) that this is ultimately owing to alterations in the balance of signaling downstream of Rac and Rho subfamily GTPase which are evoked by altered RCP-dependent recycling.

Next, the effect of mutant p53-derived exosomes is analyzed on recycling. The rationale to look at recycling is however not obvious as one might expect that a variety of different processes could be targeted and altered by these exosomes and some explanation should be provided. In Figure 2, it is, in particular, shown that exosomes derived from mutant p53 cells impair alpha5beta1 and cMet recycling in a DGKalpha dependent manner. Is the recycling of other cargo also perturbed? What about other trafficking routes (e.g. internalization)? Stated differently, it is unclear why exosomes should specifically or exclusively target RCP recycling routes. Importantly, information as to nature, quality and features (including EM or AFM analysis) of exosomes should be provided.

This has been addressed as follows:

We have measured receptor internalisation in the presence and absence of exosomes from p53^{-/-} and mutant p53-expressing cells, and have found this not to be altered (Fig. S3a). As recommended by this reviewer, we have now also extended our analysis to include the transferrin receptor, which is commonly considered to be a generic marker of recycling pathways. It is clear from these analyses that exosomes from mutant p53-expressing cells strongly promote recycling of the transferrin receptor in both H1299 cells (Fig. 2a) and fibroblasts (Fig. 5a). These data indicate that exosomes from mutant p53-expressing cells have a stimulatory influence on receptor recycling that is not necessarily restricted to RCP-dependent pathways. We have toned-down statements in the discussion to reflect this observation.

Are the mutant p53 cells-derived exosomes generally capable of impacting on recycling of alpha5beta1 and cMET in other cell lines and specifically on A2780? It is also unclear why increase recycling of different cargo (integrin and cMET) should impact on migration persistence and velocity. Can this effect entirely be understood in terms of increased FA turnover? If so, this should be directly shown?

This has been addressed as follows:

We have now measured the recycling of alpha5beta1 integrin and the transferrin receptor in A2780 cells. Indeed, exosomes from mutant p53-expressing A2780 cells increase the recycling of both alpha5beta1 and the transferrin receptor in A2780 cells (Fig. S3b).

The impact of RAB35 on the ability of mutant p53 exosomes to modulate migratory behaviors of recipient cells is interesting. However, it is unclear whether mutant p53 affect RAB35 expression and more

importantly its function? In other terms, what is the relation between mutant p53 and RAB35? Rab35 has been shown to reduced exosome secretion (Hsu et al. JCB 2010). The author here show, instead, that the number and size of exosomes after RAB35 silencing is not altered. What is the reason of this apparent inconsistency with previously published data? Please do restore the expression of RAB35 with a siRNA resistant variant or use multiple independent siRNA oligos to assess its impact of this GTPase on exosome-mediated alteration in migratory properties. The authors also showed that RAB27a and b did not oppose the ability of mutant p53 cell supernatant to perturb migration. One would expect exosome number to be severely affected after this treatment and this should be directly tested. Additionally, if exosome from RAB27a and B KD cells are not affected, then they should not be capable of altering migration as opposed to exosome derived from control mutant p53 cells.

This has been addressed as follows:

We have now undertaken a detailed mechanistic analysis of the role played by Rab35 in the intracellular sorting PODXL and how this influences its packaging into exosomes. We have now confirmed previous data from the Echard lab (Pasteur Institute) and found that Rab35 interacts physically with PODXL and controls its trafficking. We report that Rab35 is required to transport PODXL toward the plasma membrane thus reducing the amount of PODXL available to be delivered to CD63-positive late endosomes and to be packaged into exosomes. Consistently, knockdown of Rab35 opposes the migration-promoting capacity of exosomes because it diverts PODXL from the plasma membrane and leads to increased packaging of PODXL into exosomes – thus mimicking the situation in p53^{-/-} cells. We have confirmed these results by using a mutant of PODXL which is unable to interact with Rab35 and found that this mutant is packaged into exosomes far more efficiently than wild-type PODXL. By contrast, Rab27 knockdown appears to exert minimal influence on quantity of exosomes released by H1299, and has no effect on the sorting of PODXL into these structures. These data are now presented in Fig. 4.

Through Silac-proteomic and RNA-seq, it is shown that PDX is severely reduced in cells and exosomes of p53 mutant cells, but not in p53^{-/-} cells. What is the level of PDX in wt p53 cells and their exosomes? In addition, it is implied subsequently that the interaction between RAB35 and PDX in co-ip is at the basis of the suppression of migratory properties of mutant p53 cell –derived conditioned media seen after RAB35 silencing. However, it is entirely unclear whether RAB35 loss reduces cells as well exosomal PDX and how this might be occurring. Also, the set of experiment in Figure 4e and S4c suggest that PDX removal might be sufficient to generate exosomes capable of influencing migratory behavior of recipient cells regardless of p53 mutation status. Is this the case? In other word. is the reduced amount of PDX in exosome sufficient for the exosomal-mediated migratory effects? It should be tested whether exosome purified from wt, p53 mutant, p53^{-/-} in combination with PDX silencing or not are capable of perturbing migration of recipient cells and the recycling of cargos.

This has been addressed as follows:

Part of the response to this query lies in the mechanistic analysis of the role played by Rab35 in PODXL sorting, and we have covered this in the response to this reviewer's previous point (see above).

We have also investigated the consequences of inducing the expression of wild-type p53 on PODXL levels and the ability of exosomes from these cell to affect the migration of recipient cells. We have utilised H1299 cells which express wild-type p53 under an doxycycline-inducible promoter (p53-tetON). We find that induction of wild-type p53 expression does not influence PODXL levels (Fig. S5b), nor does it imbue the exosomes collected from these cells with the ability to promote cell migration of recipient cells (Fig. 2c).

To address the third component of this point, we have looked now conducted a more in-depth analysis of the role played by exosomal PODXL in evoking aspects of mutant p53's gain-of-function in recipient cells. From this it is clear that either increasing or decreasing the levels of PODXL in exosomes from mutant p53 expressing cells reduces the ability of these exosomes to drive receptor recycling etc. in recipient cells (Fig. 3e,f). To pursue this in vivo, we have now shown that CRISPR-mediated knockout of PODXL in mutant p53-expressing H1299 cells ablates their ability to alter lung ECM when implanted into nude mice (Fig. 7a). Thus the relationship between exosomal PODXL levels and the generation of pro-invasive niches is not straightforward. Our interpretation of the data so far is that mutant p53 and Rab35 collude to tune the levels of exosomal PODXL into a 'goldilocks' range which is just right for influencing receptor recycling. Any manipulation which places PODXL either above or below this range (siRNA of PODXL, overexpression of PODXL, siRNA of Rab35) renders exosomes from the cells to be ineffective in driving receptor recycling.

In 5b, it is shown that mutant p53-derived exosomes influence the trafficking of RCP cargos and migration also of immortalized dermal fibroblasts. Why a scratch would assays as opposed to a random migration assays was performed? It would be relevant to assess velocity and persistence in randomly moving fibroblasts.

Also, it is stated that "we examined the invasiveness of tumour cells in fibroblast-free Matrigel plugs, we found that treatment with mp53R273H-exosomes conferred only modest and barely-significant increases in the invasiveness of p53 null cells (not shown)." It would be relevant to show this data.

This has been addressed as follows:

We have now determined the influence of exosomes from H1299-p53^{-/-} and H1299-p53^{R273H} cells on the random migration of human dermal fibroblasts. Indeed, exosomes from H1299-p53R273H cells significantly increase the random migration speed of these fibroblasts and these data are presented in Fig. S7b.

We have also presented the Matrigel plug experiments in Fig. S7a

In 5c-d, it is shown that the ECM derived from fibroblasts exposed to exosome-derived from mutant p53 cells is structurally altered with increased orthogonally oriented collagen fibers. It is unclear why the loss of long and parallel arrays should affect the speed of MDA-MB-231 cells crawling through it. Indeed, looking at the movies, it seems that ECM derived from fibroblasts exposed to p53 mutant exosomes are organized in parallel arrays that promotes directional migration and persistent locomotion. Some explanation would seem required to account for the effect reported on migration speeds. Movies where cells are tracked concomitantly with SHG analysis of collagen fibers should be performed.

This has been addressed as follows:

In order to address this point, we have performed an extensive atomic force microscopy (AFM) analysis of the physical properties of fibroblast-derived ECM. Although, we originally hypothesized that altered stiffness might be responsible for fostering increased cell migration, this is clearly not the case. Indeed, we report that there is no consistent difference in stiffness between ECM deposited by fibroblasts exposed to mp53 or p53^{-/-} exosomes. However, an AFM approach in which we measured the force necessary to remove a silica bead which had been placed on the ECM, indicated that the ECM deposited by fibroblasts exposed to mp53 exosomes is significantly less sticky than that deposited by untreated fibroblasts or fibroblasts exposed to p53^{-/-} exosomes. We then plated cells into these matrices and measured the number and area of adhesive 3D contacts formed and found that these were significantly reduced. Thus, the 'tangled', orthogonal matrix deposited by fibroblasts treated with exosomes from mp53-expressing cells is less sticky leading to fewer adhesive interactions with cells engaging with it. This is entirely consistent with the increased ability of cells to move through this ECM. These data are now presented in Fig. 5d.

Finally, it is shown that in pancreatic cancer model driven by mutant p53 the collagen organization in the lung is perturbed. While the results from this analysis are interesting their relationship to exosome released by mutant p53 is unclear (for example whether the altered collagen organization in the lungs is caused by exosome released by mutant p53 is not shown). Additionally, it is not clear why the altered ECM, which is shown to affect migratory behavior should facilitate the seeding, colonization and growth of metastatic foci at this distal site. Is the altered ECM a property seen only in Lungs? Is the collagen organization around the primary tumors expressing mutant p53 altered similar to what is seen in the lung? One would reasonably expect that the first targets of mutant p53 exosomes would be the CAFs around the primary tumor? Is this the case? If so, one alternative explanation of the augmented metastatization is that the modified collagen around the tumors might favor local spreading?

This is a key issue which we are currently investigating. However, our investigations so far indicate that the answers to these questions are not straightforward. At present, we feel that the altered ECM of the lungs (and the liver) of mice bearing KPC (as opposed to KfC or KC) tumours is more able to support the recruitment of a number of migratory cell types – including cells of the innate immune system, such as neutrophils and macrophages. Moreover, we have some preliminary evidence that components of the 'tangled' ECM is better able to support stem-like behaviour of cancer cells. Thus,

because our preliminary investigations indicate that the answer to this reviewer's query involves an extensive investigation into the immune microenvironment of metastatic target organs, cancer stem cell behaviour and the relationship between these and ECM deposition, we feel that addressing these questions lies outwith the scope of the current manuscript.

Reviewers' comments:

Reviewer #1 (Remarks to the Author):

The authors have addressed the issues raised.

Reviewer #2 (Remarks to the Author):

The response to my comments is satisfactory

Reviewer #3 (Remarks to the Author):

This very interesting, revised work addresses a number of concerns, not all.

In particular there are few issues that are less clearly clarified

1. Previously, this reviewer raised concerns as to the role of RAB35 in relationship to mutant p53 and PODXL. The authors did some work in this direction, however, it remains unclear whether mutant p53 affects RAB35 expression/activity and what is the role of RAB35 within the mutant p53-PODXL axis. The data on RAB35-PODXL might serve as a way to perturb PODXL distribution, and thus to reinforce the role of this sialomucin, but generate some confusion with respect to the role exerted by mutant p53. Specifically,

Firstly, technically I have not seen the use of more than 1 siRNA or siRNA resistant mutants (to rescue the phenotype) of RAB35 as requested earlier.

Secondly, even in the revised version while the impact of RAB35 on PODXL distribution and exosomal localization is better defined, it remains totally unclear how and whether mutant p53 impacts on RAB35-PODXL interaction and whether this latter interaction is at all relevant for mutant p53 phenotypes. This raises the issues as to whether the RAB35 data should be included at all in the manuscript.

Thirdly the "goldilocks" hypothesis is not really a mechanistic explanation of the role played by exosomal-PODXL on recycling, it simply put in other words some of the findings. Indeed, how exosomal-PODXL impact on recycling in general (as it also affects TfR recycling) remain totally obscure.

2. The second issue pertains the impact of mutant p53 on lung stroma at distal sites. While the data provided are somewhat consistent with this possibility, it remains likely that mutant p53 is likely to impact on the organization of the stroma in the primary tumors (see also ref# 23 Miller, B.W. et al. *Embo Mol Med.* 2015 cited by the authors) in turn influencing dissemination. It would seem straight forward to assess the stromal, collagen organization not only in the lung but also in the primary tumor in their mouse PDAC models. The link between the finding obtained using cell biological approaches and the in vivo data remains somewhat weak.

The authors in the discussion try to address some of this shortcoming, as exemplified below (extracted from the discussion)

"Mutant p53-expressing tumours can influence collagen organisation in the tumour stroma and this is associated with ECM cross-linking and assembly of parallel arrays of collagen fibres. Thus, our findings clearly indicate that ECM alterations evoked at some distance from the primary tumour by mutant p53 exosomes are distinct from those observed in the primary tumour, and this suggests that this process generates a pro-invasive niche in the lungs rather than contributing to cell dissemination from the primary tumour."

However, it is not clear at all how the finding reported in the present paper "clearly indicate that ECM alterations evoked at some distance ... are distinct from those observed in the primary tumors" To support this statement the primary tumor ECM should be looked at.

3. Finally, the observation that the orthogonal ECM caused by mutant p53 exosome is "less sticky" is interesting, but why this would be, is totally mysterious. What does less sticky imply? Less

integrin binding surfaces, a different distribution/density of integrin binding sites? Why the orthogonally oriented fibers would lead to reduced affinity of interaction? Why should this lead to cells migrating on longer and straight trajectories?

Having said all this, a number of the findings reported are original and unexpected and as such interesting. It might be sufficient to perform few additional indicated experiments and reword the interpretation of some of the results obtained for this work to be published .

Reviewer 3

1. Firstly, technically I have not seen the use of more than 1 siRNA or siRNA resistant mutants (to rescue the phenotype) of RAB35 as requested earlier.

We have addressed this as follows:

To address this point, we have recapitulated experiments studying the role of Rab35 in PODXL trafficking (contained within Fig. 4) using an additional siRNA oligonucleotide sequence and, in some cases we have also used CRISPR to suppress Rab35 levels. We have now used SMARTPool siRNAs (siRNA-sp), Rab35 CRISPR cells and an additional siRNA sequence (siRNA#1) to show that Rab35 is required to traffic PODXL to the plasma membrane and these data are now included in Fig. 4b and S6b,c). We have also used an additional siRNA sequence to show that suppression of Rab35 leads to increased sorting of PODXL to CD63-positive late endosomes (Fig. 4d) and to exosomes (Fig. 4c).

1. Secondly, even in the revised version while the impact of RAB35 on PODXL distribution and exosomal localization is better defined, it remains totally unclear how and whether mutant p53 impacts on RAB35-PODXL interaction and whether this latter interaction is at all relevant for mutant p53 phenotypes. This raises the issues as to whether the RAB35 data should be included at all in the manuscript.

We have addressed this as follows:

To clarify this issue we have determined coimmunoprecipitation of PODXL with Rab35 in both p53^{-/-} and mutp53-expressing cells. These data indicate that, although mutp53 suppresses PODXL levels (as described in Fig. 3), the remaining PODXL still coimmunoprecipitates with Rab35 in mutp53-expressing cells. These data are now presented in Fig. 4a, and we have clarified the description of these data to clearly indicate that PODXL coimmunoprecipitates with Rab35 to an 'extent that is commensurate with the expression levels of PODXL in p53^{-/-} and mutp53-expressing cells respectively'. We surmise from this that the principle role of mutp53 is to (via p63) suppress PODXL expression and not to interfere with Rab35-PODXL association.

1. Thirdly the "goldilocks" hypothesis is not really a mechanistic explanation of the role played by exosomal-PODXL on recycling, it simply put in other words some of the findings. Indeed, how exosomal-PODXL impact on recycling in general (as it also affects TfR recycling) remain totally obscure.

We have addressed this as follows:

We have elucidated two distinct mechanisms through which exosomal PODXL levels may be controlled. The first is through mutp53's ability (via p63) to suppress expression of PODXL mRNA. The second, is via a direct association of Rab35 with PODXL which

influences its intracellular sorting. It is clear from our data that interference with either of these mechanisms (PODXL expression levels and PODXL sorting) reduces the ability of exosomes from mutp53-expressing cells to drive recycling in recipient cells.

We agree with this referee that we have not elucidated how exosomal PODXL influences receptor recycling. We are trying to address how this occurs and, as stated in our discussion, we are currently developing ways to manipulate PODXL glycosylation to determine whether the charge that PODXL can impart to exosomes is important to their ability to influence trafficking.

2. The second issue pertains the impact of mutant p53 on lung stroma at distal sites. While the data provided are somewhat consistent with this possibility, it remains likely that mutant p53 is likely to impact on the organization of the stroma in the primary tumors (see also ref# 23 Miller, B.W. et al. *Embo Mol Med.* 2015 cited by the authors) in turn influencing dissemination. It would seem straight forward to assess the stromal, collagen organization not only in the lung but also in the primary tumor in their mouse PDAC models. The link between the finding obtained using cell biological approaches and the in vivo data remains somewhat weak.

The authors in the discussion try to address some of this shortcoming, as exemplified below (extracted from the discussion)

“Mutant p53-expressing tumours can influence collagen organisation in the tumour stroma and this is associated with ECM cross-linking and assembly of parallel arrays of collagen fibres. Thus, our findings clearly indicate that ECM alterations evoked at some distance from the primary tumour by mutp53 exosomes are distinct from those observed in the primary tumour, and this suggests that this process generates a pro-invasive niche in the lungs rather than contributing to cell dissemination from the primary tumour.”

However, it is not clear at all how the finding reported in the present paper “clearly indicate that ECM alterations evoked at some distance ... are distinct from those observed in the primary tumors” To support this statement the primary tumor ECM should be looked at.

We have addressed this as follows:

We have used second harmonic generation/grey level correlation matrix (SHG/GLCM) analysis to quantify the differences in the organisation of the stromal ECM between mutant p53-expressing (KP^{172C}) and p53^{-/-} (KPlC) pancreatic adenocarcinoma (PDAC). These data clearly indicate that the stromal collagen in mutp53-expressing tumours is organised into more parallel arrays than found in p53^{-/-} tumours, and this is evidenced by a significant increase in the mean decay distance of collagen fibres in these tumours. This provides evidence that ‘ECM alterations evoked at some distance from the primary tumour by mutp53 exosomes are distinct from those observed in the primary tumour’ and these data are now presented in Fig. S7d.

3. Finally, the observation that the orthogonal ECM caused by mutant p53 exosome is “less sticky” is interesting, but why this would be, is totally mysterious. What does less sticky implies? Less integrin binding surfaces, a different distribution/density of integrin binding sites? Why the orthogonally oriented fibers would lead to reduced affinity of interaction? Why should this lead to cells migrating on longer and straight trajectories?

We respond to this as follows:

So far we have shown that the orthogonal ECM caused by mutp53 exosomes is physically (as determine by atomic force microscopy) less adhesive, and cells interacting with this have reduced propensity to assemble paxillin-containing (and presumably integrin-containing) adhesions. Broadly-speaking, it is thought that there is a bell-shaped relationship between adhesion and cell migration speed. Indeed, there is often an optimum level of adhesion for maximal migration and cells that are more or less adherent than this will migrate more slowly. Currently, the reason for this is thought to be that adhesions turn-over more slowly on highly adhesive substrates. But there are other possible explanations for these type of phenomena, and we are currently exploring these. We have found that mutp53 exosome-induced ECM has altered association with collagens that promote cell movement and proliferation. Thus, this issue is not straightforward and investigation of the propensity for ECM to harbour and release motogenic and mitogenic matrikines will form the basis of further investigation. We have added a statement to the discussion to illustrate that we feel this investigation lies outwith the scope of the current manuscript; Page 15 ‘..... to influence the organisation and adhesive properties of the ECM that they deposit. Although further work will be necessary to determine how this more orthogonal and less adhesive ECM influences cell migration and invasion, this type of ECM organisation, which may be clearly detected in the lungs of mutant p53 tumour-bearing animals, is more conducive to the invasive and metastatic seeding behaviour of tumour cells.’

REVIEWERS' COMMENTS:

Reviewer #3 (Remarks to the Author):

The authors have either clarified various points or when needed tone down some of the conclusions. As acknowledged there are some mechanistic work to be done, yet the manuscript is well done and the authors should be commended for addressing all the issues raised.

one minor comment:

The authors showed that the collagen organization in the primary tumors of mutant p53 PDAC is different from p53 null tumors as in the former collagen fibers are more linearly organized. This might contribute to increase metastatic propensity along with changes in the architectural organization of the ECM at the metastatic niche.

Hence the statement, in the discussion, reported in brackets below, should be slightly modified.

" Thus, our findings clearly indicate that ECM alterations evoked at some distance from the primary tumour by mutp53 exosomes are distinct from those observed in the primary tumour,

(and this suggests that this process generates a pro-invasive niche in the lungs rather than contributing to cell dissemination from the primary tumour.)

Reviewer 3

The authors have either clarified various points or when needed tone down some of the conclusions. As acknowledged there are some mechanistic work to be done, yet the manuscript is well done and the authors should be commended for addressing all the issues raised.

one minor comment:

The authors showed that the collagen organization in the primary tumors of mutant p53 PDAC is different from p53 null tumors as in the former collagen fibers are more linearly organized. This might contribute to increase metastatic propensity along with changes in the architectural organization of the ECM at the metastatic niche.

Hence the statement, in the discussion, reported in brackets below, should be slightly modified.

" Thus, our findings clearly indicate that ECM alterations evoked at some distance from the primary tumour by mutp53 exosomes are distinct from those observed in the primary tumour,

(and this suggests that this process generates a pro-invasive niche in the lungs rather than contributing to cell dissemination from the primary tumour.)

We have addressed this as follows:

We have modified this statement in the discussion as recommended by this reviewer.